# Engineering metal-carbide hydrogen traps in steels

Pang-Yu Liu [1,2,12], Boning Zhang[3,4,12], Ranming Niu [1,2,12], Shao-Lun Lu [1,5], Chao Huang[1,2], Maoqiu Wang[6], Fuyang Tian[7], Yong Mao[4], Tong Li[8], Patrick A. Burr [9], Hongzhou Lu [10], Aimin Guo[10], Hung-Wei Yen [5,11]✉, Julie M. Cairney [1,2]✉, Hao Chen [3]✉ & Yi-Sheng Chen [1,2,5]✉

Hydrogen embrittlement reduces the durability of the structural steels required for the hydrogen economy. Understanding how hydrogen interacts with the materials plays a crucial role in managing the embrittlement problems. Theoretical models have indicated that carbon vacancies in metal carbide precipitates are effective hydrogen traps in steels. Increasing the number of carbon vacancies in individual metal carbides is important since the overall hydrogen trapping capacity can be leveraged by introducing abundant metal carbides in steels. To verify this concept, we compare a reference steel containing titanium carbides (TiCs), which lack carbon vacancies, with an experimental steel added with molybdenum (Mo), which form Ti-Mo carbides comprising more carbon vacancies than TiCs. We employ theoretical and experimental techniques to examine the hydrogen trapping behavior of the carbides, demonstrating adding Mo alters the hydrogen trapping mechanism, enabling hydrogen to access carbon vacancy traps within the carbides, leading to an increase in trapping capacity.

A significant challenge to delivering hydrogen energy at scale is hydrogen embrittlement[1,2]. This problem can cause unpredictable failures of metallic components, adding significant costs for monitoring and maintaining infrastructural integrity[3]. Steel is a cost-effective material for constructing large-scale infrastructure, despite its susceptibility to hydrogen embrittlement[3]. Four methods have been considered to manage hydrogen embrittlement in steels[4]. The first is pre-service hydrogen desorption to remove pre-existing hydrogen in materials[5], but this method is not suitable for most services where hydrogen pickup occurs during service, such as bearing[6] and gas

pipes[7]. The second is adding hydrogen-impermeable coatings onto the steel surface to decrease the kinetics of hydrogen uptake[8], but the effectiveness of this method depends on the quality and durability of the coatings, which may have defects, or develop them in service through scratching, erosion, or corrosion. The third method is mixing passivating gases, such as oxygen, into the bulk transmitting gas to reduce the probability of hydrogen adsorption on the surface and absorption in materials[9]. This route comes with an additional cost associated with the separation of the gases before being used for applications such as fuel cells, where hydrogen purity is crucial. Finally,

[1]Australian Centre for Microscopy and Microanalysis, The University of Sydney, Sydney, NSW 2006, Australia. [2]School of Aerospace, Mechanical and Mechatronic Engineering, The University of Sydney, Sydney, NSW 2006, Australia. [3]School of Materials Science and Engineering, Tsing Hua University, Beijing 100084, China. [4]Materials Genome Institute, School of Materials and Energy, Yunnan University, Kunming 650091, China. [5]Department of Materials Science and Engineering, National Taiwan University, Taipei 10617, Taiwan. [6]Central Iron & Steel Research Institute Company Limited, Beijing 100081, China. [7]Institute for Applied Physics, University of Science and Technology Beijing, Beijing 100083, China. [8]Institute for Materials, Ruhr-Universität Bochum, Bochum 44801, Germany. [9]School of Mechanical and Manufacturing Engineering, University of New South Wales, Sydney, NSW 2052, Australia. [10]CITIC Metal Co., Beijing 100027, China. [11]Advanced Research Center For Green Materials Science and Technology, National Taiwan University, Taipei City, Taiwan. [12]These authors contributed equally: Pang-Yu Liu, Boning Zhang, Ranming Niu. ✉e-mail: homer.yen@ntu.edu.tw; julie.cairney@sydney.edu.au; hao.chen@mail.tsinghua.edu.cn; yi-sheng.chen@sydney.edu.au

hydrogen traps can be introduced into the material microstructure to immobilize or trap absorbed hydrogen, limiting the amount of hydrogen participating in the embrittling process[10–12]. This approach provides materials with intrinsic resistance to hydrogen embrittlement and can be applied synergistically in parallel with other approaches[4], given other application requirements such as strength, ductility, corrosion resistance, and thermal stability are not compromised[4].

To design a steel microstructure that can effectively trap hydrogen and decrease embrittlement, three criteria must be considered: 1) the susceptibility of incorporated traps to hydrogen embrittlement, 2) trap density, which determines the efficiency of the microstructure in hindering hydrogen diffusion, and 3) trapping energy, which measures how effectively a trap can retain hydrogen atoms[4,13]. Microstructural features in steel lattice, such as vacancies, dislocations, grain boundaries, and second phases (both their bulk and their interfaces) can act as hydrogen traps[4,14]. However, not all of these traps are effective in resisting hydrogen embrittlement. For instance, dislocation and vacancy traps can be produced in high density but are weak traps and can contribute to hydrogen embrittlement in steels[15–20]. Hydrogen can also be trapped at grain boundaries and phase boundaries. However, there has been suspicion around whether such trapped hydrogen under high loads can cause hydrogen-enhanced decohesion, resulting in macroscale embrittlement[18,21,22]. In addition, under high loads, hydrogen can be trapped at the crack tip where load is concentrated and lattice is expanded, forming a quasi-hydride phase that impedes the emission of dislocations, causing macroscopic cleavage in consequence[23]. So far, it is understood that the exact cause of hydrogen embrittlement is subject to many factors of material service conditions (such as applied load and hydrogen content) and hydrogen-susceptible microstructure (such as inclusions). High caution is always required when associating hydrogen-induced failures with a single mechanism. In some specific cases, it is even found that multiple mechanisms can operate synergically and simultaneously[24]. Developing a universally applicable mitigation strategy has thus been a challenging task.

Second phase precipitates, like carbides, can be strong traps, can contribute to material strength, and can be produced in high density[4]. In body-centered cubic (BCC) ferritic steels, transition metal carbides are of interest for hydrogen trapping as they can be produced at exceptionally high density with excellent dispersion. For this purpose, a metallurgic phenomenon called 'interphase precipitation' is useful to form high-density carbides in steels. This involves isothermal tempering between 600 and 800 °C immediately after austenization (i.e., without quenching prior to tempering)[25–27], so that precipitates are generated on the interphase boundary during the austenite-to-ferrite phase transformation, leading to the formation of a high density of nanosized metal carbides aligned in bands. As the density and spacing between the resulting metal carbides are sensitive to the isothermal temperatures used[27,28], it is possible to achieve engineering steel microstructures with a desired hydrogen trap density to achieve good embrittlement resistance. Furthermore, since the transition metals have low solubility in ferrite, the production of high-density metal carbides generally requires only a modest amount of alloying (typically at 0.1 to 0.3 weight percent, wt.%). This level of alloying is described as 'microalloying' and is a very cost-effective method for steel strengthening[29].

For the most effective trapping, it is desirable to create a high density of traps, but also to maximize the trapping energy and capacity. Two microstructural factors can influence this: metal carbide matrix-interface coherency and carbon vacancy in carbide. For coherent and semi-coherent interfaces, lower coherency of the ferrite-metal carbide interface means a higher number of dislocations, which can act as hydrogen traps, leading to a higher overall trapping capacity[30,31]. However, taking advantage of this effect requires coarsening (over-ageing) the metal carbides, which reduces the overall number carbides and the mechanical strength of the alloy. Additionally, excess hydrogen trapped at incoherent interfaces under high

loads can cause micro-cracks through hydrogen-enhanced decohesion (HEDE)[18,21,22]. Manipulation of interface coherency is therefore a double-edged sword when it comes to designing hydrogen traps to increase hydrogen embrittlement resistance.

Computational studies suggest that carbon vacancies in metal carbides are a promising hydrogen trap due to their high trapping energy compared to interface misfit dislocations[32–35]. When a carbon vacancy is located near a metal carbide surface, it can trap hydrogen near the interface, even if the interface is fully coherent[36]. However, theoretical models indicate that a high kinetic energy is required for hydrogen atoms to overcome the diffusion barrier near the metal carbide-matrix interface and reach the carbon vacancies inside[32–35]. Di Stefano et al.[33] have used first-principle calculations to demonstrate that it is possible to reduce this barrier by increasing the concentration of near-interface and interior carbon vacancies in titanium carbide (TiC) to create a diffusion path for hydrogen to reach strong carbon vacancy traps inside the metal carbide. A similar strategy was proposed in Kirchheim's work[37], which considers the near-interface carbon vacancy traps in vanadium carbide and is in agreement with other research[34,36]. The number of carbon vacancy sites is greater than the number of interface misfit dislocations, particularly so in transition metal carbides that have a high interface coherency and a low number of misfit dislocations at the interface. Therefore, a potential strategy for increasing hydrogen trapping capacity is to create a type of metal carbide with a very high density of 'accessible' and strong carbon vacancy traps to accommodate abundant hydrogen atoms per metal carbide. This strategy can be further leveraged against the high metal carbide number density achieved by interphase precipitation to potentially create a steel microstructure with exceptional hydrogen trapping capacity.

In this work, to explore the effectiveness of increasing the carbon vacancy concentration in metal carbides, we create two ferritic steels with a high-density of metal carbides: one with TiC and another with titanium-molybdenum carbides ((Ti,Mo)C). Both have a NaCl rock salt structure[38]. Mo, a VI-B element, substitutes for Ti, in the (Ti,Mo)C and, importantly, creates additional carbon vacancies to maintain the electron valency balance within the metal carbide lattice. We then compare the hydrogen trapping behavior of each type of metal carbide at multiple length scales. At the atomic level, we use first-principle calculations based on density functional theory (DFT) to examine the hydrogen inward migration and the presence of excess energetically stable carbon vacancies in the carbides containing Mo. At the macroscale, we employ thermal desorption spectroscopy (TDS) to measure the increased amount of trapped hydrogen in the (Ti,Mo)C steel compared to the TiC steel. The TDS specimens were intentionally left at room temperature for a long period of time to desorb the hydrogen trapped by lattice defects and to leave only carbide-trapped hydrogen for measurement. At the nanoscale, we verify that both metal carbides had similar lattice and interface structures using scanning transmission electron microscopy (STEM), showing that they had a similar size, number density, and lattice structure. Finally, we use cryogenic atom probe tomography (cryo-APT) to study the hydrogen trapping mechanisms of both metal carbides in detail, a technique developed in our previous work[39,40]. Cryo-APT allows us to preserve the hydrogen distribution in charged samples by handling hydrogen-charged samples at cryogenic temperature[41]. This allows us to compare the hydrogen trapping in (Ti,Mo)C and TiC. Our findings highlight the crucial role of carbon vacancies in transition metal carbides, forming through the addition of Mo, for a remarkable increase in hydrogen trapping capacity.

## Results

### First-principle simulations of hydrogen migration, carbon vacancy stability, and hydrogen trapping in metal carbides

To test and verify the approach of using carbon vacancies to facilitate the internalization of hydrogen trapped at a metal carbide-matrix

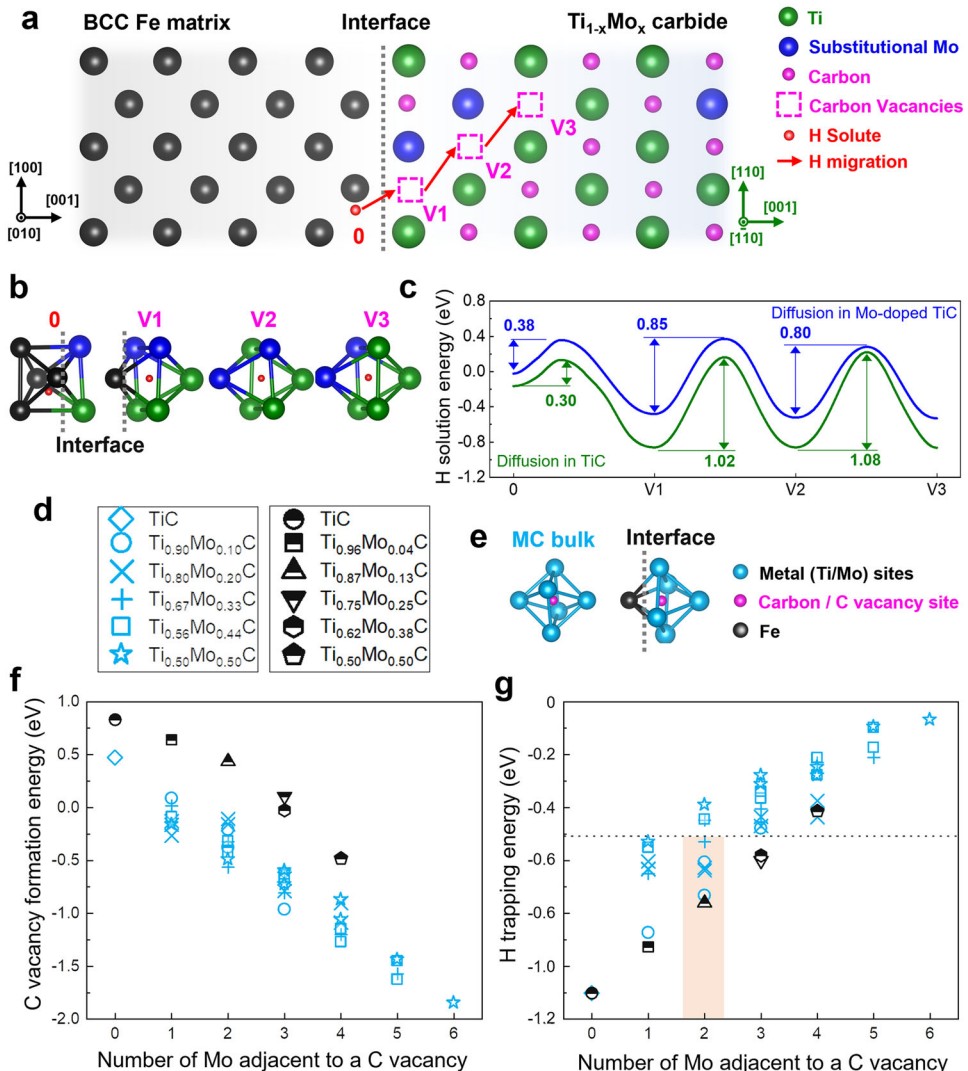

**Fig. 1 | First-principle simulations showing the effect of Mo addition on hydrogen diffusion, trapping, and carbon vacancy stability in a NaCl-structure metal carbide. a** 2-D conceptual sketch of a coherent ferrite-metal carbide interface 3-D model with 3 connected carbon vacancies (magenta dashed squares as noted V1-3), a hydrogen solute atom (red sphere at the '0' position), Fe (black spheres), Ti (green spheres), Mo (blue spheres), and C atoms (magenta spheres). The sketch is displayed along $[0\,1\,0]_{ferrite}$ which is also $[1\,1\,0]_{carbide}$ due to the orientation relationship of the two phases. **b** 3-D illustrations of the atomic coordinate configurations corresponding to the 0 and V1-3 sites in Fig. 1a. **c** Solution energy profiles corresponding to (**b**) for a hydrogen atom at the ferrite-carbide interface and in the connected carbon vacancies with (blue) and without (green)

the adjacent Mo atoms in the NaCl-structure metal carbide models. **d** Compositions and annotations of the metal carbide supercells used to create Figs. **f, g** in the case where the vacancy is inside the carbide (light blue) and at the ferrite-carbide interface (black). **e** 3-D model sketches of the carbon-site-centered unit cells considered for the TiC bulk and at the ferrite-carbide interface. **f** Energy to form a carbon vacancy as a function of the number of adjacent Mo atoms. **g** Hydrogen trapping energy in a carbon vacancy as a function of the number of adjacent Mo atoms. The dotted line at −0.5 eV, indicates the energy required to trap hydrogen for the long term in structural service at room temperature. The orange shade is the composition of the carbides in this research.

interface, we created a DFT model to probe the behavior of hydrogen at a phase boundary between BCC ferrite and a NaCl-structured metal carbide, specifically a $Ti_{1-x}Mo_x$ carbide where x represents the number of substitutional Mo atoms in a TiC unit cell. The model is depicted in Fig. 1a, which shows Fe, Ti, Mo, and C atoms in black, green, blue and magenta, respectively. Figure 1a is a 2-dimensional (2-D) view of a 3-D $(Ti_{0.75}Mo_{0.25})C_{0.875}$ supercell in the $[0\,1\,0]_{ferrite}$ direction, which corresponds to the $[1\,1\,0]_{carbide}$. This orientation relationship (OR) between ferrite and the NaCl-structure metal carbide is known as the Baker-Nutting (B−N) OR[25–27], and occurs due to the covalent bond between Fe and C atoms at the interface, leading to a highly coherent interface, as outlined by a gray dot line in Fig. 1a. The DFT model contains randomly distributed Mo atoms occupying Ti sites in the metal carbide lattice, generated using the similar atomic environment (SAE) method[42]. Three consecutive carbon vacancy sites were created

(magenta broken squares, V1-3) to simulate hydrogen diffusion toward the interior of the metal carbide at the proximity of the interface. In Fig. 1a, we placed a solute hydrogen atom (depicted as a red sphere) initially in a ferrite interstitial site near the interface, and the location of which is denoted as 0. We then analyzed the energy required for hydrogen migration (indicated by red arrows) in the presence of the three consecutive carbon vacancies.

Figure 1b is a group of 3-D models showing various sites: 0, V1, V2, and V3, with the hydrogen solute and surrounding Fe, Ti, and Mo atoms. Site 0 is a tetrahedral interstitial site in a BCC lattice near a B−N OR interface. V1 is a quasi-octahedral carbon-centered unit in a NaCl lattice near the interface, and V2 and V3 are both octahedral carbon-centered units in a NaCl lattice. We placed two Mo atoms adjacent to the vacancy, so the compositions of the models can be consistent with the bulk chemistry, i.e., $(Ti_{0.75}Mo_{0.25})C_{0.875}$. Similar models were

created (not shown) to study the hydrogen migration in TiC counterpart structures. To establish the energy profile of the hydrogen atom migrating in these models, we used climbing image nudged elastic band (CI-NEB) modeling[43] and the results are shown in Fig. 1c. This figure shows the solution energy profiles of the hydrogen atom at 0, V1, V2, and V3 sites and their corresponding saddle points. We examined the cases of a Mo-doped TiC lattice (blue) and a pure TiC lattice (green), both of which have connected carbon vacancy structures. We found that the addition of Mo reduces the energy barrier for the hydrogen atom to diffuse from one carbon vacancy site to another (0.85 eV and 0.8 eV with Mo compared to 1.02 eV and 1.08 eV for Mo-free TiC), while the carbide-entering energy barrier is only slightly increased by 0.08 eV for the Mo-containing carbide (0.38 eV versus 0.30 eV). We consider this difference is insignificant given that the diffusion energy of interstitial hydrogen in BCC iron lattice is 0.088 eV[44].

Note that our model only considered single hydrogen occupancy in each carbon vacancy. As per Di Stefano et al.[33], the diffusion barrier for a hydrogen atom decreases when a second hydrogen is present in the same carbon vacancy due to the intrinsic repulsion between ionic hydrogen particles. Their finding is particularly relevant to situations with a substantial hydrogen supply that is the regime in the hydrogen trapping observation experiments presented in later discussion.

As the quantity of carbon vacancies plays a crucial role in allowing hydrogen to access traps inside metal carbides, we used DFT to examine the effect of Mo addition on both the formation of carbon vacancies and the hydrogen trapping energy. By using SAE, we developed 6 random 3 ×3 x 3 metal carbide bulk supercells, each containing 108 metal (either Ti or Mo) sites and 108 carbon sites. The supercells for the interface models are illustrated in Supplementary Fig. 1. The compositions of both bulk metal carbide and metal carbide-ferrite interfaces are listed in Fig. 1d, and the octahedral unit cells are shown in Fig. 1e as parts of the entire supercells, where metal, carbon, and Fe sites are presented in light blue, magenta, and black, respectively. These result in the light blue and black data points, respectively, in Fig. 1f, g. We studied the carbon vacancy formation energy by removing carbon atoms in the models that have differing numbers of adjacent Mo atoms around a carbon site. As a result, Fig. 1f provides the carbon vacancy formation energy as a function of number of adjacent Mo atoms, where a more negative value indicates greater stability. Figure 1f reveals that, regardless of whether in bulk or at an interface, a carbon vacancy tends to be more stable in the presence of proximal Mo atoms, which is consistent with existing literature[45–49].

To determine the hydrogen trapping energy (i.e., the energy difference between hydrogen at traps and in a ferrite lattice), we then placed a hydrogen atom in each of the carbon vacancies, with their varying numbers of neighboring Mo atoms, resulting in Fig. 1g. We did not find that increasing the number of adjacent Mo resulted in higher trapping energy. Instead, more adjacent Mo atoms reduces the stability of the trapped hydrogen (i.e., the carbon vacancy hydrogen trapping energy), which is consistent with literature[34,50,51]. The dotted line at -0.5 eV in Fig. 1g indicates the energy below which hydrogen traps are unlikely to trap a hydrogen atom for a meaningful duration in a material service at room temperature[4]. Considering both the carbon vacancy stability and the hydrogen trapping strength, our sample with moderate Mo doping in (Ti,Mo)C, with a 1:2 Mo-to-Ti ratio (i.e., 2 Mo every 4 Ti in an octahedral cell), highlighted by the orange shade in Fig. 1g, is expected to achieve good overall synergy. Furthermore, previous research has shown that (Ti,Mo)C with this 1:2 Mo-to-Ti ratio (or $Ti_{1-x}Mo_xC$ with x = 0.33) is the most abundant carbide type in an interphase-precipitate Mo-added TiC steel, regardless of the amount of Mo addition[45]. This type of metal carbide is therefore suitable to verify the effect of incorporating a high density of carbon vacancies on the hydrogen trapping in a metal carbide-containing steel.

In summary, our modeling suggests that the addition of Mo increases the accessibility of internal carbon vacancies to hydrogen by reducing the diffusion energy barrier between the carbon vacancies in the metal carbide bulk (Fig. 1c). This diffusion barrier reduction is in addition to the formation of high-density carbon vacancies (Fig. 1f), which are likely interconnected in metal carbide bulk. Also, our model shows that the Mo-associated carbon vacancy traps have sufficient trapping strength (Fig. 1g). We consider the combination of the above can facilitate the access of hydrogen to the carbon vacancy traps in metal carbide bulk, opening up more trapping sites to the readily available carbide interface traps[52].

## Model steel design and macroscale quantification of hydrogen trapping

In accordance with the carbide design strategy described above, we used a model steel containing 0.05% carbon, 1.5% manganese, 0.2% silicon, and 0.1% titanium (by weight) and compared it to another model steel of the same composition, but with an additional 0.2% molybdenum. Both specimens have a simple microstructure with a high density of metal carbides. The moderate addition of Mo is important because excessive amounts can lead to the formation of hexagonally close-packed (HCP) $Mo_2C$ precipitates in the ferrite[28], which have lower hydrogen trapping abilities than NaCl-structure metal carbides[34].

The steels were austenitized at 1200 °C for 5 minutes to dissolve the carbide-forming alloying elements. They were then placed in an isothermal salt bath at 650 ± 20 °C for 60 minutes to form nanosized metal carbides, followed by water-quenching. Mo alloying can influence the temperatures of austenite and martensite formation and the kinetics of the austenite-ferrite phase transformation, leading to variations in the final ferrite grain size, metal carbide density, and the regularity of the metal carbide distribution[27,28]. We therefore anticipated a slight difference in microstructure, although the thermal histories of two samples are identical. Evidence of a smaller grain size of the (Ti,Mo)C steel (140 μm) than that of the TiC steel (195 μm) is provided in Supplementary Information (Supplementary Fig. 2) in the form of electron backscatter diffraction (EBSD) orientation maps and is consistent with the literature[28]. The EBSD results in Supplementary Fig. 2 also confirmed that the (Ti,Mo)C steel does not contain any detectable residual austenite that is known to be a strong hydrogen trap[4] and could cause confusion when interpreting hydrogen TDS data of carbide-containing steels.

To quantify the amount of trapped hydrogen in both steels, we carried out TDS on samples that were electrochemically charged with hydrogen. The TDS results from the TiC steel and the (Ti,Mo)C steel are shown in green and blue, respectively, in Fig. 2. After hydrogen charging, specimens were left at room temperature for either 1 or 24 hours, corresponding to the data labeled with squares or triangles, respectively, in Fig. 2. Holding at room temperature was intended to release the hydrogen that was weakly trapped in lattice defects such as dislocations and grain boundaries[4], so we can better correlate the TDS results with stronger traps such as the metal carbides in our specimens. This treatment also addresses the variance in grain size in the two specimens. To establish hydrogen-free references, we tested the hydrogen-charged specimen desorbed for 720 hours and the uncharged specimen, the corresponding data labeled as plus signs and circles, respectively, in Fig. 2. We found that (Ti,Mo)C steel (blue) preserves much larger quantities of hydrogen than TiC (green) in the steel regardless of the duration of room-temperature desorption. In the 1-hour desorption datasets (squares), the overall released hydrogen for the (Ti,Mo)C steel is 1.685 wt. ppm, which is much higher than for the TiC steel at 0.169 wt. ppm. The addition of Mo in the steel therefore increases hydrogen trapping capacity significantly. We also conducted experiments with other desorption durations, as shown in Supplementary Fig. 3 in Supplementary Information, which agrees well

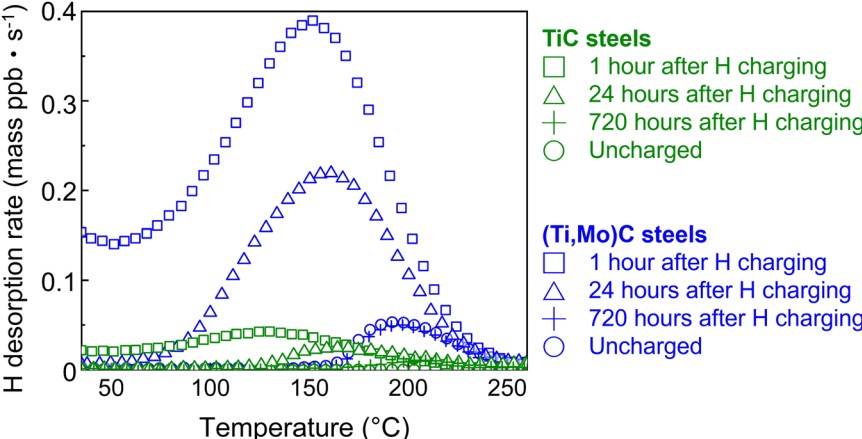

**Fig. 2 | Quantitative measurement of hydrogen trapping in the interphase-precipitate-containing ferritic steels by thermal desorption spectroscopy (TDS).** The steels with TiC (green) and (Ti,Mo)C (blue) after charging and room-temperature desorption for 1 hour (square), 24 h (triangle) and 720 h (plus sign), compared to the uncharged sample (circle).

with the data in Fig. 2. We also compared the macroscopic hydrogen permeabilities of the two specimens (Supplementary Fig. 4), revealing that the (Ti,Mo)C steel has a lower hydrogen permeability, which suggests that there are more effective hydrogen traps in the (Ti,Mo)C steel. In summary, macroscale evidence supports our hypothesis that Mo in TiC can increase hydrogen trapping capacity of steels. We then continue with microscopic analyses to reveal the origin of the enhanced hydrogen trapping.

## Microstructural characterization

We utilized bright-field TEM, atomic-resolution annular bright-field (ABF) STEM, and atomic structure simulator to examine the interphase-precipitation metal carbides in the BCC ferrite matrix, using the $[0\,1\,0]_{ferrite}$ zone axis that is most suitable for observing interphase-precipitation carbides[21–23], resulting in Fig. 3a–d from the TiC steel and Fig. 3e–h from the (Ti,Mo)Cs steel. Figures 3a, e show the TiCs and (Ti,Mo)Cs as the dark and linearly aligned features in the BCC ferrite matrix, respectively[25–27]. We examined the diffraction patterns from the TiCs and (Ti,Mo)C steels, as shown in Fig. 3b, f, confirming the presence of the diffraction spots associated with the metal carbides as highlighted by green and blue circles, respectively, as well as their B–N OR with the ferrite matrix[25–27]. We also used the metal carbides' diffraction spots to conduct dark-field imaging in TEM, as an example shown in Supplementary Fig. 5 for the TiC. However, we found this imaging method does not provide the most straightforward micrograph to exhibit the presence of metal carbides since the resulting image can also include other structural defects (Supplementary Fig. 5b).

As we aim to produce the metal carbides that have the same lattice structure (and are only different in carbon vacancy concentrations), we used ABF STEM to further confirm the atomic structure of the metal carbides and their OR with the ferrite matrix, which resulted in Fig. 3c, g. Due to the small size of the carbides relative to the specimen thickness (illustrated in Supplementary Fig. 6), the observed Ti (or substitutional Mo) atoms from the metal carbides overlap with the atomic lattice of ferrite, as highlighted by the red squares in Fig. 3c, g, hindering observations of the metal carbides. Note that the C atoms (with a low atomic mass) in the Fe matrix (with a high atomic mass) are not resolvable in our imaging condition (using 300 kV for electron acceleration). Moiré fringes, sometimes seen in conventional high-resolution TEM images of metal carbides[27,28], are absent in these STEM images because the ultra-fine scanning electron beam in STEM is less conducive to the electron beam interference that leads to Moiré fringes at the inclusions (Supplementary Fig. 6).

In order to confirm the atomic structure of the metal carbides embedded in the BCC ferrite matrix in the STEM images, we simulated the atomic structures of NaCl-like metal carbides embedded in a BCC iron matrix, as shown in Fig. 3d, h, and compared them with the STEM images (Fig. 3c, g), using the same atomic structure coordinates with respect to the imaging direction of $[0\,1\,0]_{ferrite}$ zone axis. The simulated atomic models assumed the B–N OR, i.e., $\{0\,0\,1\}_{ferrite}//\{0\,0\,1\}_{metal\ carbide}$ and $<1\,0\,0>_{ferrite}//<1\,1\,0>_{metal\ carbide}$ reported in literature[27,28]. Ti/Mo atom columns are aligned (green/blue spheres) along $(\mathbf{1\,1\,\bar{1}})$ planes in the simulated structures, as shown by the green and blue shades in Fig. 3d, h, respectively. Also, planes of Fe atom columns along $(\mathbf{1\,0\,\bar{1}})$ are highlighted by gray shades in Fig. 3d, h. Similar atomic structures in the STEM images confirm the presence of the embedded metal carbides. In the Fig. 3c, g STEM images, atom columns corresponding to the $(\mathbf{1\,1\bar{1}})_{MC}$ atom planes are highlighted by the green and blue broken lines, respectively. We also found that these $(\mathbf{1\,1\bar{1}})_{MC}$ atom columns have the same angle with respect to the $(\mathbf{1\,0\bar{1}})_{ferrite}$ planes (black broken lines in Fig. 3c and g), which is in agreement with the simulated data in Fig. 3d, h. This agreement also confirms the B–N OR between the metal carbides and the BCC ferrite matrix.

We also carried out STEM imaging along the $[1\,0\,0]_{ferrite}$ zone axis (in parallel with $[1\,1\,0]_{metal\ carbide}$ under the B–N OR), i.e., turning 90 degrees along the $[1\,0\,0]_{ferrite}$ axis toward the viewing direction of Fig. 3 (viewing from left to right in the coordinate of Fig. 3), resulting in Supplementary Fig. 7 in Supplementary Information. We found coincident locations of metal carbide and ferrite atom columns (Supplementary Fig. 7b and e), as predicted in the simulated structures in Fig. S7C, F. This is in contrast to Fig. 3b, e showing the atom columns from the $(\mathbf{1\,1\bar{1}})_{MC}$ and $(\mathbf{1\,0\bar{1}})_{ferrite}$ planes and a nominal angle between them, suggesting the observed inclusions are indeed the metal carbides with the expected NaCl crystal structure. In summary, our TEM and STEM results are consistent with previous literatures[27,28], indicating both metal carbides are NaCl type and have the B–N OR with the BCC ferrite matrix.

## High-resolution observations of hydrogen trapping in metal carbides

We then used cryo-APT to examine how hydrogen is trapped in the two metal carbides, specifically focusing on whether hydrogen atoms locate at the metal carbide-ferrite interface or within the carbide bulk, and whether this changes when additional carbon vacancies are present due to Mo in the carbides[45–49]. We utilized a cryogenic sample transfer protocol developed from a previous study[40]. Using cryo-transfer in APT analyses of hydrogen is crucial because the small

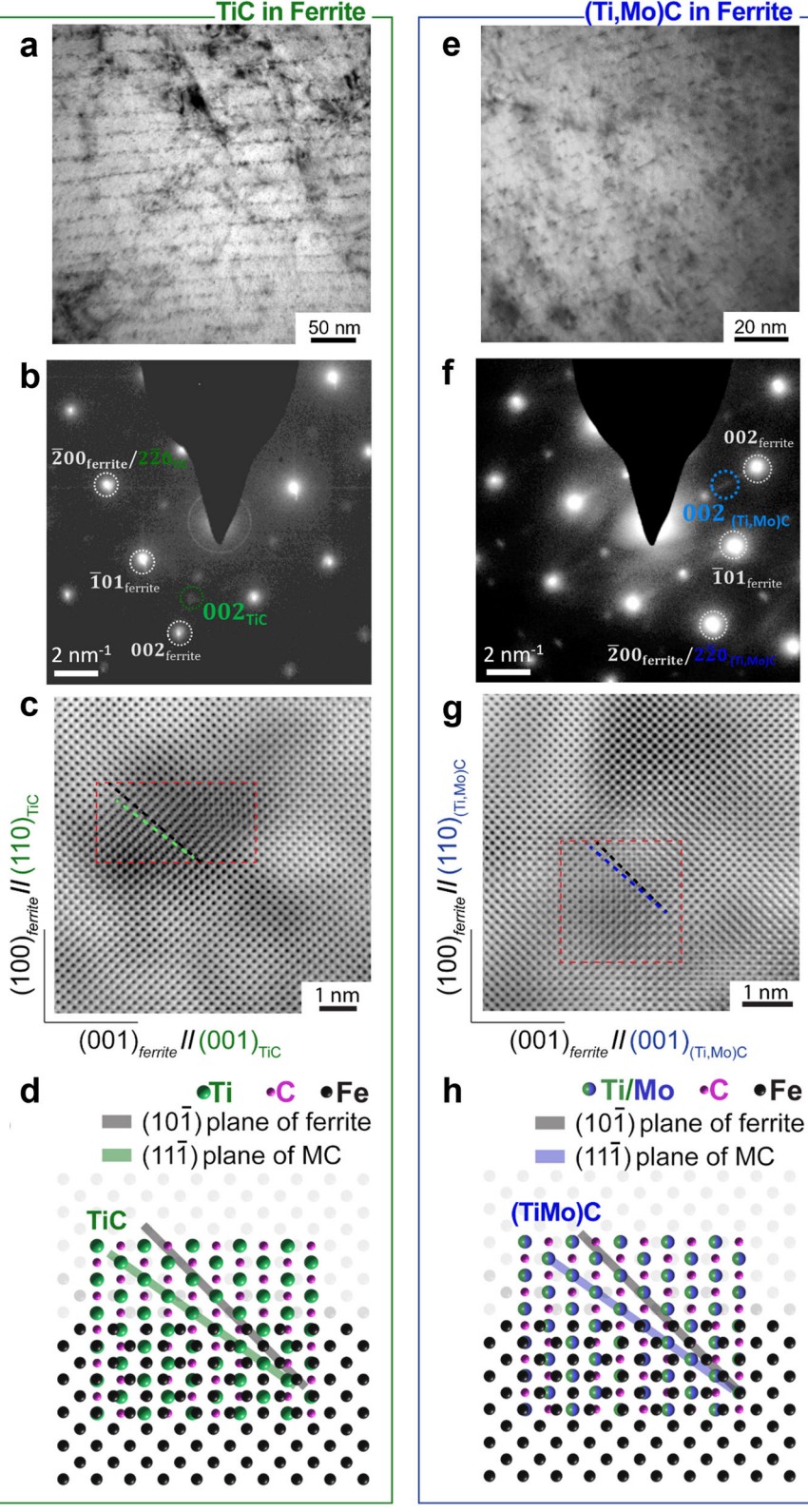

**Fig. 3 | TEM and STEM observations of the carbide morphology, structure, and orientation relationship with ferrite matrix along [0 1 0]$_{ferrite}$ zone axis. a–d** are from the TiC steel, and (**e–h**) are from the (Ti,Mo)C steel. **a** and **e** are bright-field TEM images showing the morphology of the carbides (dark and linearly aligned features). **b** and **f** are the diffraction patterns of both steel specimens, which are indexed with the spots associated with the ferrite matrix and the metal carbides. **c** and **g** are the high-resolution annular bright-field STEM images, showing the atomic structures of the NaCl-structure carbides and the BCC ferrite matrix indexes. The $(11\bar{1})$ planes of the carbides were marked by green and blue broken lines in **c** and **g**, respectively, whereas $(10\bar{1})$ planes of ferrite were marked by black broken lines. **d** and **h** are corresponding atomic models illustrating the corresponding atomic structures in the observations of **c** and **g**, showing the nominal atomic structures of the metal carbides embedded in the BCC ferrite matrix in the STEM images. These results are in consistent with previous literature [27,28], indicating the metal carbides have a NaCl structure and the B−N OR with the BCC ferrite matrix.

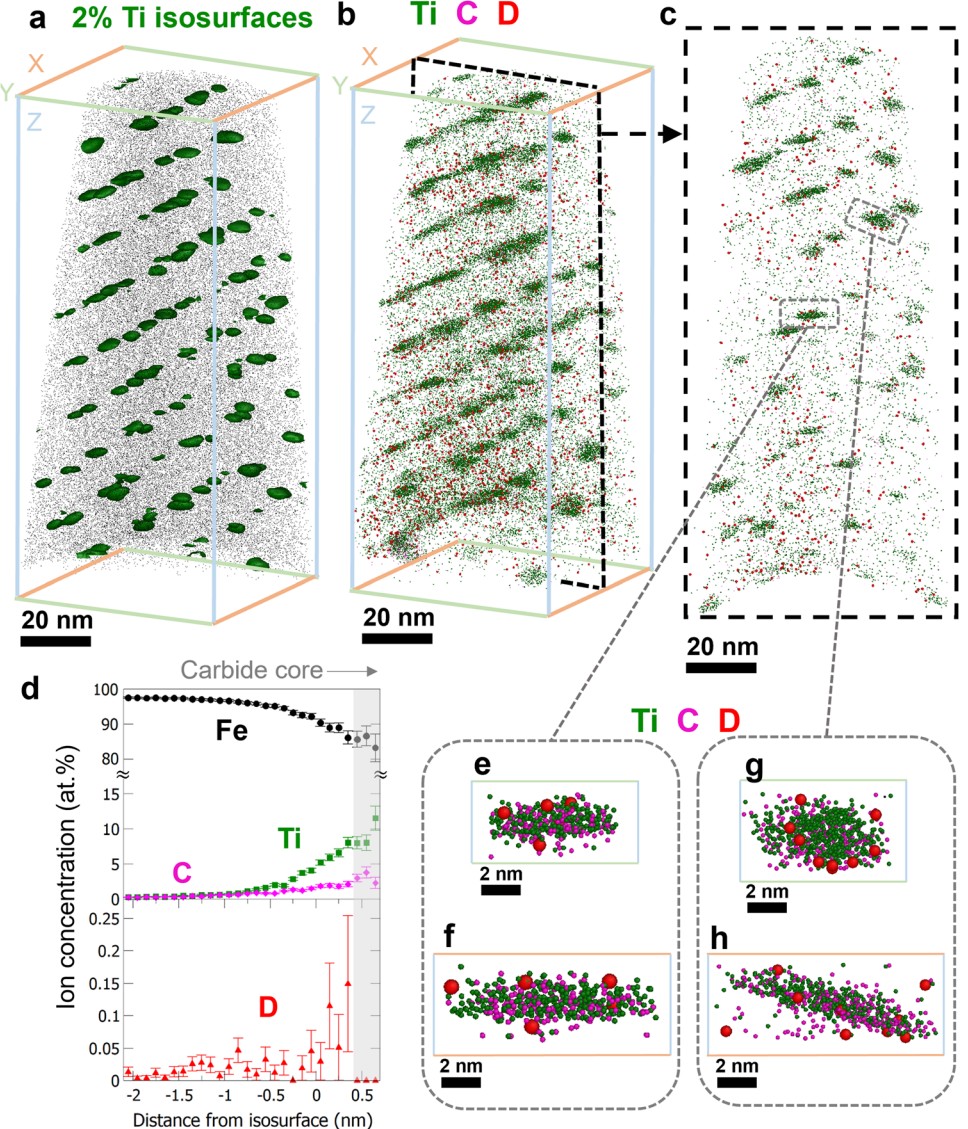

**Fig. 4 | APT analysis of deuterium-charged TiC steel. a** Reconstructed APT data showing the TiC distribution, highlighted by 2 at. % Ti isosurfaces. **b** Atom map with Ti (green), C (magenta), and D (red) displayed. **c** A 10-nm-thick slice from the broken-line-highlighted region in panel **b**. **d** An integrated proxigram from all the isosurfaces with complete shapes in **a**, showing the concentration change as a function of the distance from the defined isosurfaces. Error bars depict the confidence interval associated with each measurement of elemental composition. **e–h** Magnified views of two carbides selected from **c**, each from two perspectives.

needle-shaped samples have a very high surface-area-to-volume ratio and are susceptible to hydrogen desorption due to the high diffusivity of hydrogen in steels[53,54]. We charged our samples with deuterium (D or $^2$H), instead of normal hydrogen that contains mostly protium ($^1$H). This allows us to circumvent any ambiguity around the origin of hydrogen signal in the APT data, distinguishing the hydrogen within the sample from noise induced by the residual hydrogen in the APT ultra-high-vacuum chamber[39,40,54–56]. We used electrolytic D charging (i.e., electrolysis of heavy water), which allows a significant dose of D to maximize its signal. Details of the D charging can be found in Supplementary Information.

The APT 3-D reconstructed data of the deuterium-charged TiC and (Ti,Mo)C containing steels are shown in Figs. 4 and 5, respectively. The corresponding APT mass spectra, their D-free counterparts, and the ion peak assignments are provided in Supplementary Figs. 8 and 9. Figure 4a is a visualization of the reconstructed tip-shaped sample with the interphase-precipitate TiC carbides highlighted by 2 atomic percent (at. %) Ti isoconcentration surfaces (isosurfaces)[57] shown in green. Figure 4b shows the same volume with the Ti, C, and D atoms displayed

in green, magenta, and red, respectively. To better observe the distribution of D in relation to carbide locations, we extracted a 10-nm-thick slice along the Y-Z plane, from the area highlighted by the broken line in Fig. 4b, and this data is shown in Fig. 4c.

We used the isosurfaces defined in Fig. 4a to develop a statistical overview of the D distribution around the carbides relative to the distributions of the other elements present. Figure 4d is an integrated proximity histogram (proxigram[58]), which averages all of the proxigram data from the 52 geometrically intact isosurfaces (i.e., those not located at the edge of APT reconstruction). The proxigram shows how the concentration of different elements changes as a function of the distance from the isosurfaces. In Fig. 4d the 2 at. % Ti isosurface is the zero point on the X-axis and the more positive values are closer to the carbide core. The interfacial region corresponds to a decrease in the content of Fe (black) and increases in Ti (green) and C (magenta), with plateaus at the approximate location of carbide cores (gray shade). Note that the ratio of C to Ti in the carbide core shown in Fig. 4d is not representative of the actual carbide composition for two reasons. First, the $^{48}$Ti$^{2+}$ peak overlaps with $^{12}$C$_2^+$ peak at 24 mass-to-charge ratio (Da)

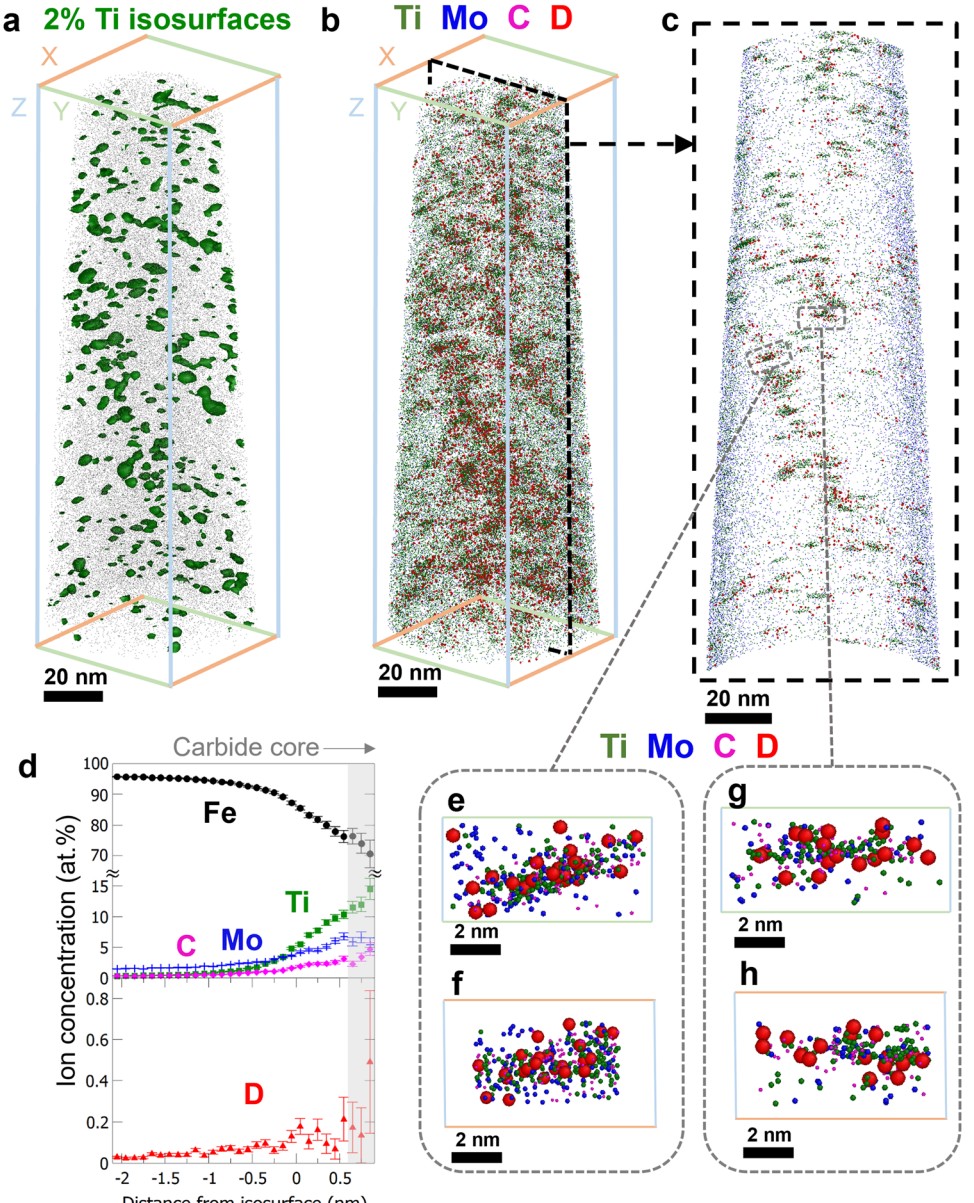

**Fig. 5 | APT analysis of deuterium-charged (Ti,Mo)C steel. a** Reconstructed data showing the (Ti,Mo)Cs highlighted by 2 at. % Ti isosurfaces. **b** Atom map with Ti (green), Mo (blue), C (magenta), and D (red) displayed. **c** A 10-nm-thick slice from the broken-line-highlighted region in panel **b**. **d** An integrated proxigram from all the isosurfaces in panel **a**. Error bars depict the confidence interval associated with each measurement of elemental composition. **e–h** Magnified views of two carbides selected from **c**, from two different perspectives.

in the APT mass spectra (Supplementary Figs. 8 and 9), impeding precise quantification of C and Ti. Second, the presence of covalently-bonded carbon atoms in carbides leads to complex field ionization and carbon-molecule dissociation in APT experiments[59–62], causing a loss of the carbon count that also affects the measured carbon concentration. Therefore, the hypo-stoichiometric C-to-Ti ratio in Fig. 4d may not relate directly to the presence of carbon vacancies in carbide core, although their presence is theoretically predicted[46–49]. We observed the expected changes in Fe, Ti, and C, as well as an increase in the D content (red) at or near the carbide surface. Interestingly, the D content drops to a very low value in the region of carbide core. The observation of higher D near the metal carbide-ferrite interface is consistent with existing literature[36], which correlates trapped D with the presence of carbon vacancy traps near the metal carbide-matrix interface. Herein, we additionally show the absence of D at the metal

carbide core, which we attribute to the inability of D atoms to access the metal carbide interior.

Proxigram analysis is useful for providing an overview of element distribution with respect to defined isosurfaces. However, the creation of proxigram involves a data voxelization process (i.e., binning considerable numbers of atoms into cubic-nanometer voxels to aggregate the atomic compositions within each voxel[58]), and this can compromise the nominal atomic resolutions in APT. To determine the hydrogen trapping sites more precisely, direct high-resolution observations of D locations were conducted, as shown in Fig. 4e–h, which are from two TiCs indicated in Fig. 4c. These atomic views show that D atoms (red) are mainly located near the metal carbide-matrix interfaces, rather than inside the metal carbide. Several previous studies have used D charging to show D atoms trapped at metal carbide interfaces[40,63], substantiating these observations. More high-resolution

atomic views of TiCs with trapped D atoms are provided in Supplementary Fig. 10a. A fly-through view of the dataset shown in Fig. 4, using 10-nm slices, is available in the online version (Supplementary Movie 1). The experiment was repeated to verify the result, as shown in Supplementary Fig. 11. These results confirm that the interface is the main hydrogen trapping site for TiC, which agrees with existing literature[54].

We then examined the hydrogen trapping in (Ti,Mo)C, as shown in Fig. 5a, and obtained an overview of the carbide distribution in the sample with the same 2 at.% Ti isosurfaces (green) used for Fig. 4a. The atom map with Mo (blue), Ti (green), C (magenta), and D (red) is shown in Fig. 5b with a 10-nm slice extracted from the region highlighted by the broken line in Fig. 5c. The (Ti,Mo)C precipitates were slightly smaller than the TiCs in Fig. 4, which is consistent with previous research that used TEM and carbon extraction replica to determine carbide sizes[45]. To gain an overview of the elemental distributions around the interface and carbide core, we again created an integrated proxigram that included the statistics of 260 isosurfaces with round geometries (Fig. 5a), which is shown in Fig. 5d. The Mo-to-Ti ratio was measured as approximately 0.5, which is consistent with previous research[45]. We observed a high concentration of D atoms (red) within the carbide core (gray shade), unlike for the TiC shown in Fig. 4d, in which hydrogen trapping occurs only at the interface.

We took a close look at the carbides with trapped D, and the views of two representative carbides are displayed in Fig. 5e, h, which are from two axial perspectives. Both show D atoms located in the carbide bulk, where carbon vacancies can exist. Additional close-up images of the (Ti,Mo)Cs with trapped D atoms are provided in Supplementary Fig. 10b. A fly-through view of the Fig. 5 dataset, using a 10-nm slice is available in the online version (Supplementary Movie 2). A repeat experiment was conducted to verify these observations in relation to the (Ti,Mo)C steel, shown in Supplementary Fig. 12. These observations provide evidence that the core of (Ti,Mo)C is the main trapping site of hydrogen, unlike that for TiC. Moreover, we observed a large difference in overall D composition between the TiC and (Ti,Mo)C steel datasets, at 0.006 at.% (Fig. 4 dataset, TiC) and 0.02 at.% (Fig. 5 dataset, (Ti,Mo)C), which is consistent with the TDS result in Fig. 2.

We also calculated the carbide number density in each APT dataset (Table 1). The carbide number densities in the two steels are of the same order of magnitude, indicating that the more trapped hydrogen in the (Ti,Mo)C steel is not simply the result of a higher carbide number density. Combining with the moderate difference in grain boundary density (Supplementary Fig. 2), the increased hydrogen trapping capacity of the (Ti,Mo)C steel is therefore attributed to the ability of the carbides to incorporate hydrogen in their bulk.

Considering the limited spatial resolution that proxigrams can achieve with voxelised data, we also examined D composition near the carbides by using a different statistical approach. For consistency between the two specimens, which have different average carbide sizes, we only compared carbides with diameters smaller than 5 nm. This helps to minimize any errors that could arise from trajectory aberrations in APT experiments, which would affect smaller carbides to a greater extent. These carbides have high interface coherency, so using the smaller precipitates for this analysis also minimizes the contribution of hydrogen trapped by misfit dislocations at the interface[30], allowing a more direct comparison of the hydrogen trapped by carbon vacancies. Figure 6a shows the carbide size distribution for TiC and (Ti,Mo)C, corresponding to the datasets in Figs. 4 and 5, respectively. The carbide sizes were defined using the Cluster Analysis algorithm in the commercial APT software (APSuite 6.0, CAMECA)[58], and the cluster-defining parameters were selected based on a protocol from the literature[64], described in Supplementary Information. The defined clusters are displayed in

Supplementary Fig. 13a, b for the TiC steel and (Ti,Mo)C steel, respectively. We used the largest radius of gyration ($R_g$) among the axes to define the radius of carbide, and the carbide size distributions in Fig. 6a are consistent with previous research using similar materials[45].

We created integrated 1-D atomic concentration profiles along the Z-axis from a centered cylindric region for all considered carbides with trapped D atoms, as conceptually illustrated in Fig. 6b. Analyses along the Z axis allow us to use the direction with the highest spatial resolution in APT, thereby reducing ambiguity when defining the locations of carbide core and interface[58]. Using the cylindric regions of interest across the carbide center allows us to exclude D atoms trapped at the lateral edges of carbide (Fig. 6b), which could be miscounted as being located in carbide core if the entire carbide is incorporated in the cylindric measurement. After collecting individual concentration profiles with a measuring step size of 1 nm, we normalized the profiles to their Ti spatial span (i.e., their size), and then superimposed the dimensionless profiles into the overall 1-D Z-axis profiles as shown in Fig. 6c, d.

Figure 6c, from the TiC APT data, shows a drop in D in the carbide core region, consistent with the proxigram data, again suggesting that TiCs are unable to trap hydrogen in their core. In contrast, Fig. 6d, from the (Ti,Mo)C steel, shows D is present in the carbide core, which is attributed to accessible carbon vacancies leading to the higher hydrogen trapping capacity, also seen in the TDS data (Fig. 2). Figure 6d also shows that the Mo-to-Ti ratio is approximately 0.5, consistent with the proxigram analysis (Fig. 5d) and previous research[45]. Although carbon quantification in APT is difficult, we believe the detected carbon contents in the two APT experiments using the identical instrument, parameters, and ion ranging method are still qualitatively comparable. In Fig. 6d, we found that the ratio of carbon content to metal (i.e., Ti plus Mo) content at the carbide core is lower than that of carbon content to Ti content in Fig. 6c. This result supports our prediction of more carbon vacancies in (Ti,Mo)C than TiC. We note that it is also possible that changes in the phase-transformation kinetics of metal carbide formation from Mo additions could also affect the carbon content. Further research to quantify the vacancies may provide a better quantitative correlation between hydrogen trapping with carbon vacancy concentration.

In summary, we have used first-principle simulations and state-of-the-art microscopic tools to demonstrate a successful materials engineering alloy design strategy to enhance the ability of steels to trap hydrogen. This is achieved by increasing the number of carbon vacancy traps that are accessible within carbide precipitates. Our microscopic observations agree well with the macroscopic analyses by thermal desorption analysis and permeation tests. This strategy can be easily implemented through microalloying and is not limited to Mo; other carbide-forming elements in steel are likely to have the same effect. This microalloying is cost-effective and applicable for a broad range of precipitate-strengthened steels.

## Methods
### First-principles calculations
To construct the interface and bulk models for $Ti_{1-x}Mo_x$ carbide, the similar atomic environment (SAE) method[42] that generates quasi-random structures through calculating binary and ternary clusters was used. Different concentrations of alloying atoms ranging from 0.125 to 0.50 in the Ti sub-lattice were selected according to chosen number of atoms in supercells. The calculation of H trapping energetics is consistent with that in[50]. The formation energy of C vacancies in various sites was calculated according to[65]:

$$\Delta E_{C-Vac} = E^{C-Vac}_{system} - E_{system} + \mu_C \qquad (1)$$

**Table 1 | Carbide number density from APT analyses**

| Sample | Related figure | Data volume ($10^{-22}$ m³) | Carbide number* | Number density ($10^{21}$ m³) |
|---|---|---|---|---|
| TiC steel | Fig. 4 | 8.47 | 129 | 1.52 |
| | Supplementary Fig. 11 | 10.98 | 293 | 2.67 |
| (Ti,Mo)C steel | Fig. 5 | 9.11 | 284 | 3.12 |
| | Data not shown | 14.46 | 164 | 1.13 |

*Carbides were defined by using the protocol developed in[64].

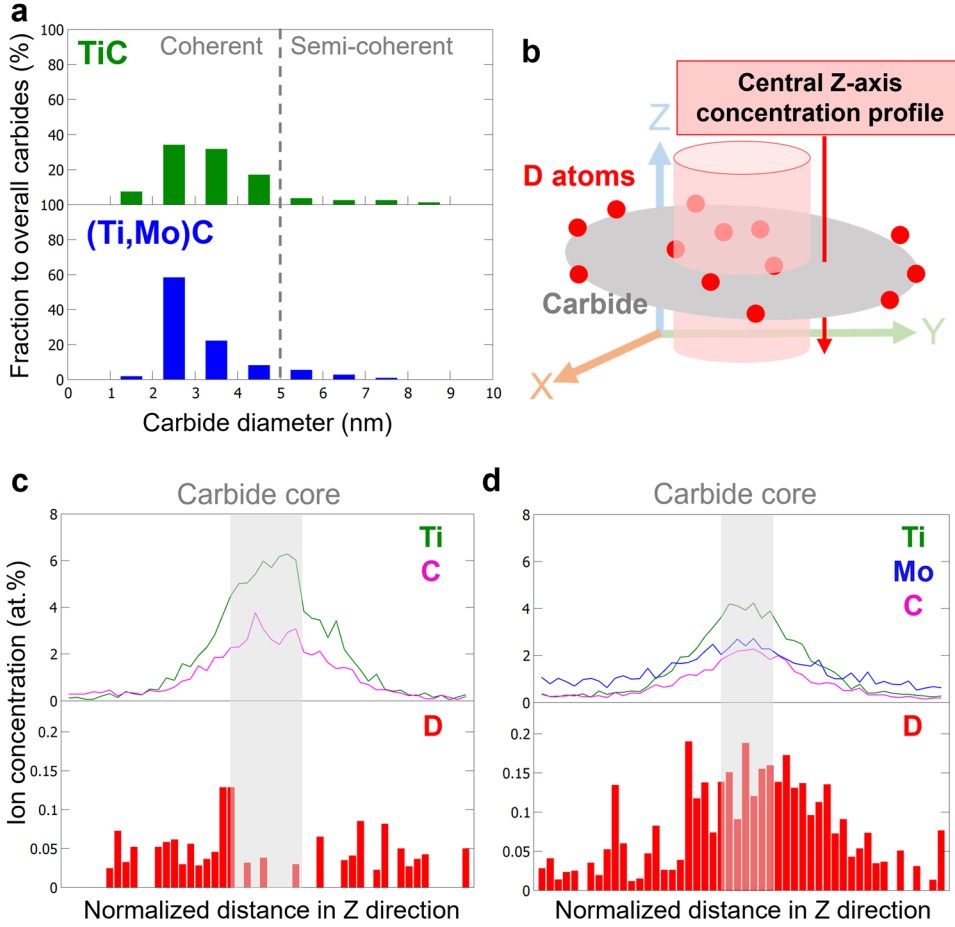

**Fig. 6 | Statistical analyses of deuterium distributions in relation to interface and carbide core locations from Figs. 4 and 5 results. a** presents the carbide size histograms of TiC (green) and (Ti,Mo)C (blue) from the Figs. 4 and 5 datasets, respectively. **b** is a schematic illustration of the method to extract a 1-D Z-axis concentration profile from the center of a carbide in APT dataset. 1-nm step size was used to generate the profiles. **c** is the overall Z-axis profile from the TiC steels shown in Fig. 4 after normalizing each profile with corresponding carbide size and superimposing all the resulting dimensionless profiles. **d** is the counterpart of **c** for (Ti,Mo)C from Fig. 5.

where $E_{system}^{C-Vac}$ ($E_{system}$) is the total energy of the system with (without) a C vacancy, $\mu_C$ is the referenced chemical potential of C, which is equal to the single-atom energy of graphite. The formula describes the off-stoichiometric tendency in the carbides under C-rich conditions.

Ab-initio calculations based on DFT were carried out using the VASP code[66]. The projector augmented wave (PAW) potentials and the generalized gradient approximation (GGA)[67] with the Perdew-Burke-Ernzerhof (PBE)[68] parametrization of exchange-correlation functionals were adopted. A plane-wave cutoff energy of 450 eV and Monkhorst-Pack[69] k-point sampling was used for atomic relaxations until the forces are smaller than 0.01 eV/Å. The climbing image nudged elastic band (CI-NEB) method[43] was used to find the energy barrier of H migration, where five images were used with the relaxation criterion set at 0.05 eV/Å.

## Scanning electron microscopy and electron backscatter diffraction experiments

The steel samples were polished using SiC polishing papers of 30, 15, 5, and 1 μ grades, followed by a final 1-minute polishing in the Struers Inc TegraPol-25 instrument using active oxide polishing suspensions. The surface of the samples was then cleaned using Ar ion beam polishing with a Gatan PECS ll machine, using two 2 kV beams with an inclination angle of ± 3°. EBSD experiments were conducted in a Thermofisher G4 Hydra Plasma FIB-SEM, prior to preparing TEM samples using the FIB lift-out method. During the EBSD experiments, a 20 kV 1.8 nA electron beam was used, and EBSD patterns were collected using an Oxford Instruments Symmetry S1 EBSD Detector. The Kikuchi patterns were indexed using both the 'fcc iron' and 'bcc iron' phases in the embedded database in the EBSD software Aztec. The data presented in

Supplementary Fig. 2 has an index rate of more than 95%, which were all automatically indexed to the bcc iron phase, indicating that no austenite was detected.

### Thermal desorption spectroscopy and hydrogen permeation tests

Mechanically polished specimens with diameters of 5 mm and length of 25 mm were electrochemically H-charged using an aqueous solution of 3% NaCl + 0.3% $NH_4SCN$ with a current of 0.3 mA/cm$^2$ for 48 h. The H-charged specimens were then exposed at ambient temperature for selected durations from 0 to 720 h before conducting TDS measurements using the HTDS-002 instrument at Central Iron & Steel Research Institute Company. To verify that Mo alloying affects H diffusion through the nano-precipitates rather than the substituted atoms in the ferrite matrix, electro-chemical H permeation tests were conducted on the model steels and compared with that of a commercially pure Fe alloyed with 0.5 wt.% Mo (Fe-0.5Mo). The tests were carried out at a constant potential of 300 mV in an aqueous solution of 0.2 M/L NaOH + 0.3 wt.% $NH_4SCN$.

### Transmission electron microscopy

A Thermo-Fisher G4 Hydra Plasma FIB-SEM with EBSD detector was used to conduct TEM and STEM lift-out sample preparations. Grains with the $[100]_{ferrite}$ crystal orientation were targeted to lift-out 15-micron-thick samples. These lift-out samples were placed on copper grids and thinned by a xenon plasma ion beam, in the sequence of 30 kV-1nA, 30 kV-100 pA, 30 kV-30 pA, and finally 5 kV-30pA. 3 lift-out samples from 3 different grains (with the same crystal orientation) were prepared and examined to confirm the reproducibility of STEM data.

TEM analyses were conducted on a FEI Tecnai G2 F20. STEM analyses were conducted on a Spectra 300 kV STEM and a Thermo-Fisher Themis-Z 300 kV STEM with two aberration correctors for low-resolution and atomic-resolution imaging, respectively. The samples were mounted onto double-tilted holders and were cleaned in Ar plasma before entering the STEM. We imaged the samples in the $<001>_{ferrite}$ zone axis. We used the electron probe aperture of 25.1 mrad in semi-angle and the ABF collection semi-angles between 48 and 200 mrad. We used a monochromator to set the probe current at 80 pA, in order to minimize the electron beam damage.

### Atom probe tomography and specimen deuteration

The charging and cryo-transfer methods were adopted from our previous work[40]. We conduct the charging in a custom-designed cryo-transfer glove box. The charging condition for the sample is 30 seconds with 2.2 V, followed by the liquid nitrogen bath and the cryo-transfer process. A more comprehensive description of the charging and cryo-transfer method is reported in the supporting document of[39].

For the sample preparation, the steels were cut into 1 × 1 × 15 mm matchstick-shaped bars. We first use the 25% perchloric acid in acetic acid at 10-30 V for rough electropolishing, separating the sample from the middle and yielding two needle-shaped samples. This is followed by fine electropolishing with 2% perchloric acid in butoxyethanol under a 40x optical microscope. The voltage range for the fine polishing is around 5-20 V; the higher voltage is used to form the necking and the lower voltage to remove the top part. All the atom probe experiments were analyzed with a Local Electrode Atom Probe (LEAP 4000 Si, CAMECA) with a pulse frequency of 200 kHz, a stage temperature of 50 K and a pulse fraction of 20%. After acquiring the data, the reconstruction was conducted by the atom probe analysis software AP Suite (version 6.1, CAMECA). For the reconstruction parameters, the detector efficiency of 57% and image compression factor 1.40 were used for TiC specimen and 1.44 for (Ti,Mo)C specimen in the APT data reconstructions.

## Data availability

All data can be accessed online via Github with Zenodo: https://doi.org/10.5281/zenodo.10416768[70].

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

## Acknowledgements

Y.-S. C. thanks for the support from his family. The authors acknowledge funding from Australian Research Council (LP180100431, LP210300999, IE230100160, FT180100232, and LE190100048), CITIC-CBMM Nb Steel Award Fund Program (2018FWNB30064), the 2019 University of Sydney Postdoctoral Fellowship, Taiwan's Ministry of Education (Taiwan-University of Sydney Scholarship) and National Science and Technology Council (111-2119-M-002 –020-MBK), China's National Key R&D Program (2022YFE0110800 and 2022YFB3705200) and National Natural Science Foundation (51922054, 52301176, and U1808208), Advanced Computing Center of Yunnan University, National Taiwan University, as well as the research facilities and technical service supported by Microscopy Australia and the University of Sydney's Core Research Facility program.

## Author contributions

P.-Y.L. conducted the APT experiments and the data analyses and drafted the manuscript. B. Z. conducted the DFT simulation, the TDS experiments, and drafted the manuscript. R.N. conducted the TEM experiments and drafted the manuscript. S.-L.L. and C.H. supported the APT and TEM experiments. M.W. supported the TDS experiments. F.T. supported the DFT simulation. Y.M. provided the computational resource for the simulation and supported the DFT simulation. T.L. conceptualized the project and participated in the manuscript development. P.A.B. participated in the data analysis and the manuscript development. H.L., A.G., and H.-W.Y. provided funding support, supervised the project, and participated in the manuscript development. J.M.C. provided the microscope facilities, supervised the project, and finalized the manuscript. H.C. initiated and conceptualized the project, provided the samples and funding, supervised the project, and participated in the manuscript development. Y.-S.C. initiated and conceptualized the project, designed the protocols of analyses, provided funding support, supervised the project, drafted the manuscript, and finalized the manuscript.

## Competing interests

The authors declare no competing interests.
