## [Peer Review File · Nature Communications]

Engineering metal-carbide hydrogen traps in steelsREVIEWER COMMENTS

Reviewer #1 (Remarks to the Author):

This is an excellent, carefully executed paper on the effect of vacancy concentration in carbides on the trapping of hydrogen (deuterium). I believe it should be published but have some recommendations for improvement:

The description of hydrogen embrittlement mechanisms is very selective. While I appreciated this is a large, and controversial, field, and the paper is not concerned with embrittlement per se, I do think this selectively is too narrow. There are interesting recent critiques of the mechanisms (e.g. Song & Curtin, *Nature Mater.*, 12 (2013) 145-151). These other mechanisms are important because of the role that trapped hydrogen plays. The HELP mechanisms requires only diffusible hydrogen and hydrogen trapped at dislocations/grain boundaries, which is always present, irrespective of the deep traps in carbides etc.

P5: "MC matrix-interface incoherency and carbon vacancy in carbide". It would be better to say interface coherency. Most of the carbides in microstructures generated by interphase precipitation are coherent/ semi-coherent.

P5: "Additionally, excess hydrogen trapped at incoherent interfaces under high loads can cause hydrogen-enhanced decohesion (HEDE), leading to micro-crack initiation at the interface and resulting in macroscale embrittlement". Yes, this maybe true (although Song and Curtin calculate that HEDE cannot be a feasible mechanism), but even hydrogen trapped at coherent interfaces can be released in a stress field.

Last line introduction "Our theoretical and experimental results provide complementary evidence to demonstrate a new microstructural design strategy that can be easily and cost-effectively implemented for the development of hydrogen-compatible steels with exceptional hydrogen trapping capabilities." This is a really important point. Firstly, this is not new as the references cited in the paper show. Secondly, just because the precipitates are better at trapping does not mean that the steel is more resistant to hydrogen embrittlement. The authors should look at Gong et al. (*Acta Materialia*, 223(1) (2021) 117488), who investigated an almost identical Mo containing TiC interphase precipitated steel and found that it was severely embrittled by hydrogen. Interestingly, the vanadium variant was not, with the V₄C₃ probably containing more vacancies than the Ti(Mo)C, with the V₄C₃ certainly trapping hydrogen, as shown in the cited Science paper. Therefore, the premise that "more trapping inside the carbide is better" is not necessarily true. The important point is whether in the stress field of a propagating crack that trapped hydrogen remains trapped or is released, with the released hydrogen leading to further crack decohesion. This needs much more careful wording in the document- some might say that the basic approach in this paper is flawed as it has been shown to not work.

"We found that the addition of Mo reduces the energy barrier for the hydrogen atom to diffuse from one carbon vacancy site to another (0.85 eV and 0.8 eV with Mo compared to 1.02 eV and 1.08 eV for Mo-free TiC), while the carbide-entering energy barrier is slightly higher for the Mo-containing carbide (0.38 eV versus 0.30 eV)." This statement is incompatible with the following statement "In short, our modelling suggests that the addition of Mo reduces the barrier for inward diffusion of hydrogen in the presence of connected carbon vacancies in MCs." The latter statement does not support the results in Fig. 1 unless there are multiple hydrogen atoms in the vacancy, which is far from proven. As the calculations show, the solution energy is higher with Mo present than without. A much more careful discussion is required here. Also, I think the calculations of Restrepo et al. (*Intl. J. Hydrogen Energy*, 45 (2020) 2382) are highly relevant here; while they looked at VC and not TiC, the structures are essentially the same and the issues around vacancies are the same and importantly, the barrier by the interface to hydrogen solution in the carbide is a really important issue.

Reviewer #2 (Remarks to the Author):

The objective of this work is the design of an alloy with a density of defects likely to trap hydrogen irreversibly. The authors used DFT calculations to compare several configurations of carbides with different concentrations of carbon vacancies. Their calculations showed that the addition of molybdenum (replacing titanium) would promote the formation of carbon vacancies, and therefore the irreversible trapping of hydrogen. Then, the authors elaborated two microstructures, one with titanium carbide (TiC) and the other with an addition of molybdenum (Ti, Mo)C, then characterised the atomic structures of the carbides and identified the relationship of orientation with the Fe matrix (same atomic structures). Following this, hydrogen charging and TDS dosing tests showed an increase in the concentration of H in the microstructure with (Ti, Mo)C carbides compared to that with TiC carbides. More in-depth characterisations were conducted by Cryo-APT on samples pre-charged with deuterium seem to confirm greater irreversible trapping by the precipitates (Ti, Mo)C, which would validate the hypothesis of trapping by carbon vacancies, and therefore the objective of designing a new alloy capable of irreversibly trapping hydrogen and more resistant to the hydrogen embrittlement (HE).

It is a significant and impressive work, it required the mobilisation of several skills (theoretical and experimental) and advanced experimental technics. The results obtained are very interesting and bring a new approach to developing alloys more resistant to HE.

This work is based on two ideas which are:

I. Substitution of Ti atom by Mo in TiC carbide is accompanied by an increase in the concentration of carbon vacancies. This stabilises the growth of the carbide and the crystalline structure and maintains the coherency of the carbides with the matrix.

II. Carbon vacancies are irreversible traps for hydrogen.

These two approaches have been largely demonstrated by DFT (and other calculations), and experimentally using several technics (HRTEM, APT, TDS...), which question about the scientific originality of this work. Here is a non-exhaustive list of references that attest to this:

1. <https://doi.org/10.1016/j.matchemphys.2021.125178>
2. <https://doi.org/10.1016/j.actamat.2011.09.051>
3. <https://doi.org/10.1002/pssb.2221940214>
4. <https://doi.org/10.1016/j.actamat.2018.05.003>
5. <https://doi.org/10.1016/j.actamat.2014.04.051>
6. DOI: 10.1039/C5CP07724A
7. V. V. KRAINIK et al.: The Electronic Structure of Mo,Ti, -C, Carbide Alloys, *phys. stat. sol. (b)* 194, 575 (1996)
8. <https://doi.org/10.1016/j.ijhydene.2023.07.264>
9. Wei, F.G., Tsuzaki, K. Quantitative analysis on hydrogen trapping of TiC particles in steel. *Metall Mater Trans A* 37, 331–353 (2006). <https://doi.org/10.1007/s11661-006-0004-3>
10. <https://doi.org/10.1016/j.corsci.2020.108929>
11. 10.1038/s41467-022-31665-x (the same team)
12. <https://doi.org/10.1016/j.matdes.2022.110399>
13. Seol, J.-B. et al. Core-shell nanoparticle arrays double the strength of steel. *Sci. Rep.* 7, 42547; doi: 10.1038/srep42547 (2017).

On the other hand, this work raises several points that need to be clarified:

- 1) The consequences of the increase in the concentration of carbon vacancies on the mechanical

behavior and the functional properties of the alloys?

2) The carbon vacancies can be at the origin of the formation of nanovoids in the presence of H?

3) Hydrogen promotes the formation of vacancies SAV, which would lead to the formation of clusters of vacancies and even nanovoids.

4) By modifying the chemistry of the alloy, it is possible to modify the surface reactivity, therefore the domains of the hydrogen evolution reaction (HER), and therefore the cathodic charging conditions. But the authors apply the same conditions, it is possible that the difference in H concentration can also be linked to the charging conditions?

5) On the EBSD maps of the two alloys, I see a preferential texture: (111) for TiC steel and (101) for (Ti, Mo)C steel. In addition to this difference in texture, texture can also influence surface reactivity.

6) A significant increase in the concentration of carbon vacancies can modify the lattice parameter. But I don't see any particular effect.

7) What is the consequence of this approach on resistance to HE?

8) If the choice of molybdenum seems interesting, other works have shown that vanadium can also increase the concentration of carbon vacancies, what do the authors think

Reviewer #3 (Remarks to the Author):

This paper investigates the effect of Mo-doping of titanium carbides on hydrogen trapping in steel. This gives valuable data for answering some fundamental questions about the trapping of hydrogen, which has many ramifications for alloy design in hydrogen embrittlement. The paper is well-written and potentially of great interest to the community. However, there are a few issues which should be addressed.

1) The study is presented with the context of hydrogen trapping at carbides in steels being useful for removing hydrogen from contributing to embrittlement. While there has been some correlation to better performance in mechanical tests of steels with carbides compared to without, the long-term performance is unclear, as hydrogen traps can be saturated with hydrogen and then their efficacy is unknown. Especially since this paper does not provide any mechanical data showing that the better trapping of the engineered carbides actually improves the performance in hydrogen, a little more tempered statement of the motivation, acknowledging there are other subtleties to the issue, might be appropriate.

2) The page 4 line 111 statement "transition metal carbides (MC, with M as Ti, V, Mo, Nb, etc.) are of interest" is potentially confusing as written. "MC" can be used as an abbreviation for the term metal carbides, however it is also a potential carbide structure/chemical formula (along with M₃C, M₂C₃, M₇C₃, etc.). Separating the list of possible transition metal species from the same parenthetical as the definition of the acronym will avoid the confusion between abbreviation and formula, while the addition of a clear statement about the structure of the carbides in this study would also help readers.

3) Given the smaller grain size of the (Ti,Mo)C steel, a part of the reduction in the hydrogen permeability (Page 13) is likely due to the increased number of grain boundaries, and the attendant trap sites, and is not solely due to the increased trapping capability of the carbides. This contribution to the permeability should be addressed in the paper.

Additionally, the difference in grain size, easily derived from the EBSD data, should be given to help give readers a tangible idea of the differences between the two alloys.

4) In discussing the differences in hydrogen trapping and permeation between the two alloys, some quantitative difference in the carbide numbers or distributions between the two alloys would be very useful. While this is partially addressed during the discussion of the atom probe data, it can be argued that knowing the microstructure is critical to understanding the trapping and permeation data. Alternatively, the trapping and permeation could be brought up again after the carbide distributions are discussed to put those results in context.

5) For Fig. 3, given that high resolution TEM images do not give structural information, unlike diffraction data, but can be interpreted with the help of atomic models, such as in this figure, the phrasing "These results confirm ..." is a little too strong. "These results are consistent with ..." is more accurate. (Similar statements occur within the text and caption for the figure.)

6) Also, for Figure 3, did the authors attempt to get diffraction patterns of the carbides, recognizing the difficulty due to their small size? Also, given the contrast in the TEM micrographs, perhaps dark field images would better show the differences in carbide distribution? Figure 3D is an especially difficult micrograph to interpret, possibly due to the FIB process to make the sample.

Reviewer #1:

This is an excellent, carefully executed paper on the effect of vacancy concentration in carbides on the trapping of hydrogen (deuterium). I believe it should be published but have some recommendations for improvement:

1) The description of hydrogen embrittlement mechanisms is very selective. While I appreciated this is a large, and controversial, field, and the paper is not concerned with embrittlement per se, I do think this selectively is too narrow. There are interesting recent critiques of the mechanisms (e.g. Song & Curtin, Nature Mater., 12 (2013) 145-151). These other mechanisms are important because of the role that trapped hydrogen plays. The HELP mechanisms requires only diffusible hydrogen and hydrogen trapped at dislocations/grain boundaries, which is always present, irrespective of the deep traps in carbides etc.

We appreciate the reviewer's suggestion and have integrated the description about the nano-hydride formation theory proposed by Song and Curtin into the hydrogen embrittlement mechanism discussion at Page 4 in the revised manuscript, which reads as below:

“Hydrogen can also be trapped at grain boundaries and phase boundaries. However, there has been suspicion around whether such trapped hydrogen under high loads can cause hydrogen-enhanced decohesion (HEDE), leading to micro-crack initiation at the interface and resulting in macroscale embrittlement^{18,21,22}. In addition, under high loads, hydrogen can be trapped at the crack tip where load is concentrated and lattice is expanded, forming a quasi-hydride phase that impedes the emission of dislocations, causing macroscopic cleavage in consequence²³. So far, it is understood that the exact cause of hydrogen embrittlement is subject to many factors of material service conditions (such as applied load and hydrogen content) and hydrogen-susceptible microstructure (such as inclusions). High caution is always required when associating hydrogen-induced failures with a single mechanism. In some specific cases, it is even found that multiple mechanisms can operate synergically and simultaneously²⁴. As such, developing a universally applicable mitigation strategy has been a challenging task..”

2) P5: “MC matrix-interface incoherency and carbon vacancy in carbide”. It would be better to say interface coherency. Most of the carbides in microstructures generated by interphase precipitation are coherent/ semi-coherent.

The text has been revised as suggested at Page 5 in the revised manuscript.

3) P5: “Additionally, excess hydrogen trapped at incoherent interfaces under high loads can cause hydrogen-enhanced decohesion (HEDE), leading to micro-crack initiation at the interface and resulting in macroscale embrittlement”. Yes, this maybe true (although Song and Curtin calculate that HEDE cannot be a feasible mechanism), but even hydrogen trapped at coherent interfaces can be released in a stress field.

We have revised the description with a less assertive statement, which reads as below at Page 4 in the revised manuscript:

“Hydrogen can also be trapped at grain boundaries and phase boundaries. However, there has been suspicion around whether such trapped hydrogen under high loads can cause hydrogen-enhanced decohesion (HEDE), leading to micro-crack initiation at the interface and resulting in macroscale embrittlement ^{18,21,22}.”

4) Last line introduction “Our theoretical and experimental results provide complementary evidence to demonstrate a new microstructural design strategy that can be easily and cost-effectively implemented for the development of hydrogen-compatible steels with exceptional hydrogen trapping capabilities.” This is a really important point. Firstly, this is not new as the references cited in the paper show. Secondly, just because the precipitates are better at trapping does not mean that the steel is more resistant to hydrogen embrittlement. The authors should look at Gong et al. (Acta Materialia, 223(1) (2021) 117488), who investigated an almost identical Mo containing TiC interphase precipitated steel and found that it was severely embrittled by hydrogen. Interestingly, the vanadium variant was not, with the V₄C₃ probably containing more vacancies than the Ti(Mo)C, with the V₄C₃ certainly trapping hydrogen, as shown in the cited Science paper. Therefore, the premise that “more trapping inside the carbide is better” is not necessarily true. The important point is whether in the stress field of a propagating crack that trapped hydrogen remains trapped or is released, with the released hydrogen leading to further crack decohesion. This needs much more careful wording in the document- some might say that the basic approach ion this paper is flawed as it has been shown to not work.

We appreciate the reviewer’s comments regarding 1) the novelty of the proposed concept of creating extra carbon vacancies in carbide precipitates to increase hydrogen trapping capacity, 2) the link between hydrogen trapping capacity and hydrogen embrittlement resistance, and 3) the question around hydrogen trapping

state in the presence of high load.

Regarding the novelty of the research idea, although the existing theoretical research has predicted carbon vacancy hydrogen trapping, as highlighted in the manuscript already, the present work is an important and unambiguous verification that substantiates the theory. This verification requires significant intellectual inputs in experiment design, sample fabrication, microscopic observation, and data analysis. Combining these approaches requires a harmonic cooperation among multiple research groups specializing in cryogenic atom probe tomography, transmission electron microscopy, first-principle simulation, steel metallurgy, and hydrogen embrittlement, respectively. These specialties require more than 10 years of development, and the presented results require 3 years of a proper delivery. None of these are easier than purely computational prediction. Having said that, we have acknowledged the pioneering modelling work and are willing to revise the statements relating to the novelty in the manuscript, with the intention to emphasize the new experimental findings we are reporting here. The last line in the introduction now reads:

“Our findings highlight the crucial role of carbon vacancies in transition metal carbides, forming through the addition of Mo, for a remarkable increase in hydrogen trapping capacity.”

Regarding the usefulness of hydrogen traps for reducing hydrogen embrittlement susceptibility, as noted in our previous response to the reviewer’s Comment 1, it is essential to consider all the critical factors when it comes to defining hydrogen embrittlement susceptibility. The known key factors include the level of applied load (with respect to material yield strength), the mode of hydrogen supply (continuous or intermittent), the intensity of hydrogen loading (resulting in the variance in trap saturation), and material microstructure (the presence of hydrogen embrittlement susceptible features such as high-angle grain boundaries where intergranular failure favor).

For the applications where hydrogen supply is finite, such as those of automotive components, a meaningful assessment of hydrogen embrittlement susceptibility requires a controlled hydrogen charging, which can introduce a defined amount of hydrogen without saturating the traps. In this regime, the capacity and number of hydrogen traps are critical for extending the component lifetime before reaching the embrittling content of hydrogen in lattice. Theoretically, Fernández-Sousa et al. has provided a clear conclusion to this argument [Analysis of the influence of microstructural traps on hydrogen assisted fatigue. Acta Materialia 199,

253-263 (2020)]. Experimentally, a work to demonstrate this strategy is by Nagao, et al. [The effect of nanosized (Ti,Mo)C precipitates on hydrogen embrittlement of tempered lath martensitic steel. *Acta Materialia* 74, 244–254 (2014)]. Another one is by Depover and Verbeken [The effect of TiC on the hydrogen induced ductility loss and trapping behavior of Fe-C-Ti alloys. *Corrosion Science* 112, 308-326 (2016)], which used the electrolytic charging applying a moderate intensity (0.8 mA/cm² for 1 h in a 0.5 M H₂SO₄) and resulted in the comparable amounts of hydrogen in the specimens with and without carbide traps for demonstrating the decrease of hydrogen embrittlement susceptibility in the presence of carbide hydrogen traps. Contrasting to the above research with moderate hydrogen charging conditions, Gong et al. [*Acta Materialia*, 223(1) (2022) 117488] (the paper was published in 2022, not 2021 as the reviewer quoted) used a strong electrolytic charging, i.e., 10 mA/cm² for at least 1 h and up to 48 h in 0.5 M NaCl aqueous solution with 0.3 wt.% NH₄SCN. This recipe can saturate the Ti-Mo carbide hydrogen traps and exaggerate the effect of hydrogen, which unsurprisingly led to more obvious embrittlement. We thus disagree with the reviewer's reference of Gong et al.'s research to disprove the usefulness of carbide hydrogen traps. We consider the research on the role of carbide hydrogen trap in the design of hydrogen embrittlement resistant steels is still in development, which requires more experimental research to establish.

Finally, the reviewer mentioned the possible release of hydrogen from the traps in the presence of high load. It is important to note that ceramic carbide precipitates tend to have much higher load resistances than the metallic matrix during crack propagation. Therefore, in a hydrogen embrittlement process, it should be better to store hydrogen atoms in the ceramic carbide bulk than at the interface or matrix where crack could encounter. However, we have not acquired sufficient evidence to distinguish the effect of carbide-stored hydrogen from that of interface-trapped hydrogen in a hydrogen-participating crack propagation. We appreciate the reviewer pointed out this aspect that we are also very keen to develop and look forward to providing more research outcomes to address the interest in this research field.

In summary, we have decoupled hydrogen trapping and hydrogen embrittlement resistance in the revised manuscript to avoid possible confusion.

5) “We found that the addition of Mo reduces the energy barrier for the hydrogen atom to diffuse from one carbon vacancy site to another (0.85 eV and 0.8 eV with Mo compared to 1.02 eV and 1.08 eV for Mo-free TiC), while the carbide-entering energy barrier is slightly higher for the Mo-containing carbide (0.38 eV versus 0.30

eV).” This statement is incompatible with the following statement “In short, our modelling suggests that the addition of Mo reduces the barrier for inward diffusion of hydrogen in the presence of connected carbon vacancies in MCs.” The latter statement does not support the results in Fig. 1 unless there are multiple hydrogen atoms in the vacancy, which is far from proven. As the calculations show, the solution energy is higher with Mo present than without. A much more careful discussion is required here. Also, I think the calculations of Restrepo et al. (Intl. J. Hydrogen Energy, 45 (2020) 2382) are highly relevant here; while they looked at VC and not TiC, the structures are essentially the same and the issues around vacancies are the same and importantly, the barrier by the interface to hydrogen solution in the carbide is a really important issue.

We thank the reviewer pointed out the statement that may cause confusion. We meant to highlight two points that can facilitate the hydrogen trapping in metal carbide bulk: a) the presence of extra carbon vacancies due to the Mo substitution to Ti in the metal carbides, as noted in the introduction at Page 7 in the revised manuscript, and b) the slight reduction of diffusion barrier between the carbon vacancies in the metal carbide bulk (i.e., 0.85 eV and 0.8 eV with Mo compared to 1.02 eV and 1.08 eV for Mo-free TiC), as demonstrated by our DFT calculation. We should also have mentioned that the 0.08 eV increase of hydrogen diffusion barrier at the interface between metal carbide and matrix is insignificant with respect to the interstitial hydrogen diffusion energy in BCC iron (0.088 eV). I.e., a hydrogen atom diffusing in BCC iron must overcome at least 0.088 eV and would not be much hindered by any energetic barrier smaller than this. Moreover, as per the findings of Di Stefano et al.³⁴, it is established that the presence of multiple hydrogen atoms in a single vacancy can reduce the trapping energy significantly. Our modelling result is consistent with their finding and adds little value to iterate on the base of Di Stefano et al. To better articulate the points above, we added more information at Page 9 in the revised manuscript:

“We found that the addition of Mo reduces the energy barrier for the hydrogen atom to diffuse from one carbon vacancy site to another (0.85 eV and 0.8 eV with Mo compared to 1.02 eV and 1.08 eV for Mo-free TiC), while the carbide-entering energy barrier is only slightly increased by 0.08 eV for the Mo-containing carbide (0.38 eV versus 0.30 eV). We consider this difference is insignificant given that the diffusion energy of interstitial hydrogen in BCC iron lattice is 0.088 eV⁴⁵.”

“Note that our model only considered single hydrogen occupancy in each carbon vacancy. As per Di Stefano et al.³³, the diffusion barrier for a hydrogen atom decreases when a second hydrogen is present in the same carbon vacancy due to the

intrinsic repulsion between ionic hydrogen particles. Their finding is particularly relevant to situations with a substantial hydrogen supply that is the regime in the hydrogen trapping observation experiments presented in later discussion.”

Additionally, to give a more wholistic summary from the DFT results, we moved the summarizing paragraph to the end of the modelling section, as at Page 11 in the revised manuscript, with the content that better clarifies our points and incorporates the reference of Restrepo et al. as suggested by the reviewer:

“In summary, our modelling suggests that the addition of Mo increases the accessibility of internal carbon vacancies to hydrogen by reducing the diffusion energy barrier between the carbon vacancies in the metal carbide bulk (Fig. 1C). This diffusion barrier reduction is in addition to the formation of high-density carbon vacancies (Fig. 1F), which are likely interconnected in metal carbide bulk. Also, our model shows that the Mo-associated carbon vacancy traps have sufficient trapping strength (Fig. 1G). We consider the combination of the above can facilitate the access of hydrogen to the carbon vacancy traps in metal carbide bulk, opening up more trapping sites to the readily available carbide interface traps.⁵³”

Reviewer #2:

The objective of this work is the design of an alloy with a density of defects likely to trap hydrogen irreversibly. The authors used DFT calculations to compare several configurations of carbides with different concentrations of carbon vacancies. Their calculations showed that the addition of molybdenum (replacing titanium) would promote the formation of carbon vacancies, and therefore the irreversible trapping of hydrogen. Then, the authors elaborated two microstructures, one with titanium carbide (TiC) and the other with an addition of molybdenum (Ti, Mo)C, then characterised the atomic structures of the carbides and identified the relationship of orientation with the Fe matrix (same atomic structures). Following this, hydrogen charging and TDS dosing tests showed an increase in the concentration of H in the microstructure with (Ti, Mo)C carbides compared to that with TiC carbides. More in-depth characterisations were conducted by Cryo-APT on samples pre-charged with deuterium seem to confirm greater irreversible trapping by the precipitates (Ti, Mo)C, which would validate the hypothesis of trapping by carbon vacancies, and therefore the objective of designing a new alloy capable of irreversibly trapping hydrogen and more resistant to the hydrogen embrittlement (HE).

It is a significant and impressive work, it required the mobilisation of several skills (theoretical and experimental) and advanced experimental technics. The results obtained are very interesting and bring a new approach to developing alloys more resistant to HE.

This work is based on two ideas which are:

- I. Substitution of Ti atom by Mo in TiC carbide is accompanied by an increase in the concentration of carbon vacancies. This stabilises the growth of the carbide and the crystalline structure and maintains the coherency of the carbides with the matrix.
- II. Carbon vacancies are irreversible traps for hydrogen.

These two approaches have been largely demonstrated by DFT (and other calculations), and experimentally using several technics (HRTEM, APT, TDS...), which question about the scientific originality of this work. Here is a non-exhaustive list of references that attest to this:

1. <https://doi.org/10.1016/j.matchemphys.2021.125178>
2. <https://doi.org/10.1016/j.actamat.2011.09.051>
3. <https://doi.org/10.1002/pssb.2221940214>
4. <https://doi.org/10.1016/j.actamat.2018.05.003>
5. <https://doi.org/10.1016/j.actamat.2014.04.051>

- 6.DOI: 10.1039/C5CP07724A
- 7.V. V. KRAINIK et al.: The Electronic Structure of Mo,Ti, -C, Carbide Alloys, phys. stat. sol. (b) 194, 575 (1996)
- 8.<https://doi.org/10.1016/j.jhydene.2023.07.264>
- 9.Wei, F.G., Tsuzaki, K. Quantitative analysis on hydrogen trapping of TiC particles in steel. Metall Mater Trans A 37, 331–353 (2006). <https://doi.org/10.1007/s11661-006-0004-3>
- 10.<https://doi.org/10.1016/j.corsci.2020.108929>
- 11.10.1038/s41467-022-31665-x (the same team)
- 12.<https://doi.org/10.1016/j.matdes.2022.110399>
- 13.Seol, J.-B. et al. Core-shell nanoparticle arrays double the strength of steel. Sci. Rep. 7, 42547; doi: 10.1038/srep42547 (2017).

We appreciate the reviewer elaborated the publications, which are mostly theoretical and worth citing in the present experiment-focused research. We have added the relevant citations at Page 6 in the revised manuscript. To iterate, although several simulation studies have suggested that carbon vacancies within metal carbides can act as hydrogen traps, our work provides the first experimental evidence to verify the hypothesis unambiguously.

On the other hand, this work raises several points that need to be clarified:

1) The consequences of the increase in the concentration of carbon vacancies on the mechanical behavior and the functional properties of the alloys?

This study aims to demonstrate the change of hydrogen trapping behavior due to the increase of carbon vacancy number in metal carbides. We consider the experimental approach, particularly the hydrogen charging condition, is only useful for hydrogen mapping and not suitable for studying hydrogen embrittlement susceptibility. In term of the change of mechanical property due to the carbon vacancy in the metal carbides, we reckon this change will not be as significant as the change of carbide number density resulting from Mo addition. Carbide number density can dictate the mean free path of dislocation, but the number of carbon vacancy in the metal carbide cannot. Although the reviewer raised an interesting question, we do not think we are able to provide meaningful answer to the question given the research approach used in the present study.

2) The carbon vacancies can be at the origin of the formation of nanovoids in the presence of H?

3) Hydrogen promotes the formation of vacancies SAV, which would lead to the formation of clusters of vacancies and even nanovoids.

To Comments 2 and 3 together:

It is important to note that the formation of vacancies in metal carbides is not a result of hydrogen charging nor the mechanical loading. It is a by-product of the transition metal carbide precipitation in the metallurgical annealing process. As such, the carbon vacancy is very different to superabundant vacancy (SAV) by Fukai and his co-workers, which refers specifically to the formation of numerous metal lattice defect vacancies as a result of hydrogen charging at an abnormally high fugacity. It is also different to the nanovoid that forms at a crack front during crack propagation. The reviewer might need to elaborate more about how the reviewer thinks these terms can be related to the carbon vacancy in metal carbide studied in the present research to facilitate a meaningful discussion.

4) By modifying the chemistry of the alloy, it is possible to modify the surface reactivity, therefore the domains of the hydrogen evolution reaction (HER), and therefore the cathodic charging conditions. But the authors apply the same conditions, it is possible that the difference in H concentration can also be linked to the charging conditions?

Generally, microalloying does not lead to significant change in electrochemical properties or surface reactivity unless the element has a strong intention for surface migration such as Cr. We have not noticed any literature reporting the effect of Mo alloying to hydrogen evolution reaction. We would love to further investigate this aspect if the reviewer can better indicate the significance.

5) On the EBSD maps of the two alloys, I see a preferential texture: (111) for TiC steel and (101) for (Ti, Mo)C steel. In addition to this difference in texture, texture can also influence surface reactivity.

As described in the Section XX, our process for material preparation involves no factors that can lead to a preferential texture. This XXX texture is simply a result of the selection of data presentation.

Our material fabrication did not involve any straining procedures that can lead to a textured microstructure. We revisited the EBSD data and confirmed there is no preferred microstructure orientation can be extracted.

6) A significant increase in the concentration of carbon vacancies can modify the lattice parameter. But I don't see any particular effect.

To iterate, the carbon vacancies in metal carbides are not a result of straining perfect carbides. The high concentration of carbon vacancy is formed during the interphase precipitation process and the specimens are free of stress and strain. As such, the change of lattice parameter would be mainly due to the substitution of Mo to Ti, which is not significant enough to be straightforwardly detected in either TEM or APT. The reviewer might need to suggest few publications relating to this effect to facilitate a meaningful discussion .

7) What is the consequence of this approach on resistance to HE?

As discussed in the response to Reviewer 1's Comment 4, the assessment of hydrogen embrittlement resistance requires a rigorous methodology to correlate with hydrogen trapping. The introduction of hydrogen traps is particularly useful when the hydrogen supply is finite, as demonstrated by Sousa et al. [Analysis of the influence of microstructural traps on hydrogen assisted fatigue. *Acta Materialia* 199, 253-263 (2020)], Nagao et al [The effect of nanosized (Ti,Mo)C precipitates on hydrogen embrittlement of tempered lath martensitic steel. *Acta Materialia* 74, 244–254 (2014)], and Depover and Verbeken [The effect of TiC on the hydrogen induced ductility loss and trapping behavior of Fe-C-Ti alloys. *Corrosion Science* 112, 308-326 (2016)].

8) If the choice of molybdenum seems interesting, other works have shown that vanadium can also increase the concentration of carbon vacancies, what do the authors think

We have highlighted the potential to generalize this microalloying approach to other elements for increasing the hydrogen trapping capacity in the as-submitted manuscript, now at the Page 29 of the revised version. We appreciate the reviewer revisiting this point and highlighting the high impact of the proposed material design strategy.

Reviewer #3:

This paper investigates the effect of Mo-doping of titanium carbides on hydrogen trapping in steel. This gives valuable data for answering some fundamental questions about the trapping of hydrogen, which has many ramifications for alloy design in hydrogen embrittlement. The paper is well-written and potentially of great interest to the community. However, there are a few issues which should be addressed.

1) The study is presented with the context of hydrogen trapping at carbides in steels being useful for removing hydrogen from contributing to embrittlement. While there has been some correlation to better performance in mechanical tests of steels with carbides compared to without, the long-term performance is unclear, as hydrogen traps can be saturated with hydrogen and then their efficacy is unknown. Especially since this paper does not provide any mechanical data showing that the better trapping of the engineered carbides actually improves the performance in hydrogen, a little more tempered statement of the motivation, acknowledging there are other subtleties to the issue, might be appropriate.

We agree with the reviewer's comment and have revised the description with less definitive wording, as our response to Reviewer 1's Comment 4.

2) The page 4 line 111 statement "transition metal carbides (MC, with M as Ti, V, Mo, Nb, etc.) are of interest" is potentially confusing as written. "MC" can be used as an abbreviation for the term metal carbides, however it is also a potential carbide structure/chemical formula (along with M₃C, M₂C₃, M₇C₃, etc.). Separating the list of possible transition metal species from the same parenthetical as the definition of the acronym will avoid the confusion between abbreviation and formula, while the addition of a clear statement about the structure of the carbides in this study would also help readers.

As suggested by the reviewer, all the 'MC' has now been replaced by 'metal carbide' in the revised manuscript.

3) Given the smaller grain size of the (Ti,Mo)C steel, a part of the reduction in the hydrogen permeability (Page 13) is likely due to the increased number of grain boundaries, and the attendant trap sites, and is not solely due to the increased trapping capability of the carbides. This contribution to the permeability should be addressed in the paper.

Additionally, the difference in grain size, easily derived from the EBSD data, should

be given to help give readers a tangible idea of the differences between the two alloys.

As suggested, we have added the information of grain size in the Supplementary Information: TiC specimen: 195 um and (Ti,Mo)C specimen: 140 um. This information clarifies that the tenfold increase of hydrogen trapping in Mo-added carbide steel must not be solely due to the increase of grain boundary density, which can lead to a threefold increase, i.e. $(190/140)^3 = 2.7$.

4) In discussing the differences in hydrogen trapping and permeation between the two alloys, some quantitative difference in the carbide numbers or distributions between the two alloys would be very useful. While this is partially addressed during the discussion of the atom probe data, it can be argued that knowing the microstructure is critical to understanding the trapping and permeation data. Alternatively, the trapping and permeation could be brought up again after the carbide distributions are discussed to put those results in context.

We agree with the reviewer's comment regarding that the APT results can give the best picture of the materials' carbide number densities. However, we intend to leave the APT data as the most critical results to present toward the end. Where is the best place to provide the information of APT carbide density with respect to the description about the macroscale characterization such as permeability is indeed tricky, and we consider the present order is better as compared to place APT results at the beginning of the result section.

Having said that, we have added a sentence in the final paragraph in Page 29 after presenting and discussing the APT data to better address the reviewer's concern:

"...number of carbon vacancy traps that are accessible within carbide precipitates. Our microscopic observations agree well with the macroscopic analyses by thermal desorption analysis and permeation tests. This strategy..."

5) For Fig. 3, given that high resolution TEM images do not give structural information, unlike diffraction data, but can be interpreted with the help of atomic models, such as in this figure, the phrasing "These results confirm ..." is a little too strong. "These results are consistent with ..." is more accurate. (Similar statements occur within the text and caption for the figure.)

As suggested, the description has been modified to the following sentence:

“These results are in consistent with previous literature^{23,24}, indicating the metal carbides have a NaCl structure and the B–N OR with the BCC ferrite matrix.”

6) Also, for Figure 3, did the authors attempt to get diffraction patterns of the carbides, recognizing the difficulty due to their small size? Also, given the contrast in the TEM micrographs, perhaps dark field images would better show the differences in carbide distribution? Figure 3D is an especially difficult micrograph to interpret, possibly due to the FIB process to make the sample.

We appreciate the reviewer’s advice on the TEM images in Figure 3. We have incorporated the new TEM images with improved image clarities and the associated diffraction patterns, as shown in Figs. 3A and 3B. , 3E, and 3F.

Figs. 3A and 3E are the bright-field TEM images. We further provide the TEM diffraction patterns from the TiC and (Ti,Mo)C steel specimens, as shown in Figs. 3B and 3F, respectively, where the diffraction spots associated with the (0 0 2) of metal carbide are highlighted in green and blue. However, as seen in the diffraction patterns, the intensity of electron scattering relating to the carbides was not strong due to their tiny sizes hence low volume fractions in the specimens. We thus consider our HR-STEM analyses are still necessary for defining the orientation relationship between carbide and matrix.

TiC in Ferrite

(Ti,Mo)C in Ferrite

We also attempted the dark-field imaging. An example dataset from the TiC specimen is shown in Fig. S5 in the Supplementary Information and as below. We found using dark-field TEM does not provide a micrograph that shows only metal carbide, i.e., the beam condition for the metal carbide can also make other defects visible, which obscures the metal carbide observation. We thus believe providing the bright-field images, in combination with the diffraction patterns and the HR TEM analyses, are most suitable for characterizing the metal carbides.

A. Bright-field image of TiC

B. Diffraction pattern

C. Dark-field image of TiC

According to the new results, we updated the content in the main text at Pages

15 and 16 as below:

“We utilized bright-field TEM, atomic-resolution annular bright-field (ABF) STEM, and atomic structure simulator to examine the interphase-precipitation metal carbides in the BCC ferrite matrix, using the [0 1 0]ferrite zone axis that is most suitable for observing interphase-precipitation carbides²¹⁻²³, resulting in Figs. 3A-D from the TiC steel and Figs. 3E-H from the (Ti,Mo)Cs steel. Figs. 3A and 3E show the TiCs and (Ti,Mo)Cs as the dark and linearly aligned features in the BCC ferrite matrix, respectively²⁵⁻²⁷. We examined the diffraction patterns from the TiCs and (Ti,Mo)C steels, as shown in Figs. 3B and 3F, confirming the presence of the diffraction spots associated with the metal carbides as highlighted by green and blue circles, respectively, as well as their B–N OR with the ferrite matrix²⁵⁻²⁷. We also used the metal carbides’ diffraction spots to conduct dark-field imaging in TEM, as an example shown in Fig. S5 for the TiC. However, we found this imaging method does not provide the most straightforward micrograph to exhibit the presence of metal carbides since the resulting image can also include other structural defects (Fig. S5B).”

REVIEWERS' COMMENTS

Reviewer #1 (Remarks to the Author):

The authors have addressed all the main issues, and provided a robust rebuttal where they disagree. I think the paper is now worthy of publication.

Reviewer #2 (Remarks to the Author):

I thank the authors for taking into account my comments and suggestions, and especially for answering my various questions. Reading the revised version and the authors' responses, I realize that this quality work has been enriched and that the results obtained by different approaches (experimental and DFT) will make it possible to better understand hydrogen trapping by carbon vacancies at the carbides and precipitates, and will be important for future development and design of new metallurgical states more resistant to hydrogen embrittlement.

Reviewer #3 (Remarks to the Author):

The authors have appropriately responded to the reviewers comments, with the slight omission that in editing the section on TEM, they removed the definition of "B-N OR" which was written out as "Baker Nutting Orientation relationship" in the earlier draft.